# Breaking spore dormancy in budding yeast transforms the cytoplasm and the solubility of the proteome

Samuel Plante[1,2,3,4,5], Kyung-Mee Moon[6], Pascale Lemieux[1,2,3,4,5], Leonard J. Foster[6], Christian R. Landry [1,2,3,4,5] *

1 Institut de Biologie Intégrative et des Systèmes (IBIS), Université Laval, Québec (Québec), Canada, 2 Regroupement Québécois de Recherche sur la Fonction, l'Ingénierie et les Applications des Protéines (PROTEO), Université Laval, Québec (Québec), Canada, 3 Département de biologie, Université Laval, Québec (Québec), Canada, 4 Département de biochimie, microbiologie et bio-informatique, Université Laval, Québec (Québec), Canada, 5 Centre de recherche en données massives (CRDM), Université Laval, Québec (Québec), Canada, 6 Department of Biochemistry & Molecular Biology, and Michael Smith Laboratories, University of British Columbia, Vancouver (British Columbia), Canada

* christian.landry@bio.ulaval.ca

## Abstract

The biophysical properties of the cytoplasm are major determinants of key cellular processes and adaptation. Many yeasts produce dormant spores that can withstand extreme conditions. We show that spores of *Saccharomyces cerevisiae* exhibit extraordinary biophysical properties, including a highly viscous and acidic cytosol. These conditions alter the solubility of more than 100 proteins such as metabolic enzymes that become more soluble as spores transit to active cell proliferation upon nutrient repletion. A key regulator of this transition is the heat shock protein, Hsp42, which shows transient solubilization and phosphorylation, and is essential for the transformation of the cytoplasm during germination. Germinating spores therefore return to growth through the dissolution of protein assemblies, orchestrated in part by Hsp42 activity. The modulation of spores' molecular properties are likely key adaptive features of their exceptional survival capacities.

## Introduction

Organisms across the tree of life rely on dormancy to withstand hostile conditions. This cellular state implies an arrest of the cell cycle and of cell metabolism and changes in cell properties that favor survival under unfavorable conditions [1,2]. For instance, nematodes, rotifers, and tardigrades produce dormant life stages that allow them to resist acute stresses such as freezing, desiccation, and heat stresses [3–5]. In flowering plants, the embryo develops as a dormant seed, which contributes to its survival over a long period of time by resisting drought and mechanical stress until it reaches favorable conditions to resume growth [6]. Cell dormancy is also an adaptive strategy in cancer cells, whereby metastatic cells become dormant after dissemination and resume proliferation after treatment has succeeded at eliminating the primary tumors [7]. As one of the most widespread adaptive

**Data Availability Statement:** The mass spectrometry proteomics data have been deposited to the ProteomeXchange Consortium via the

PRIDE partner repository with the dataset identifier PXD035403.

**Funding:** This work was supported by Natural Sciences and Engineering Research Council of Canada (NSERC) Discovery Grants to CRL (RGPIN-2020-04844) and LJF (RGPIN-2022-03022), a Canadian Institutes of Health Research (CIHR) Foundation grant (387697) to CRL, and platform funding from Genome Canada (264PRO) to LJF. CRL holds the Canada Research Chair in Cellular Synthetic and Systems Biology. The funders had no role in study design, data collection and analysis, decision to publish, or preparation of the manuscript.

**Competing interests:** The authors have declared that no competing interests exist.

**Abbreviations:** ACD, alpha-crystallin domain; AUC, area under the curve; GO, gene ontology; iBAQ, intensity-based absolute quantification; LC-MS/MS, liquid-chromatography-coupled tandem mass spectrometry; MAP, mitogen-activated protein; MSD, mean squared displacement; NTR, N-terminal region; PCA, principal component analysis; TEM, transmission electron microscopy.

survival strategies to extreme conditions, understanding the molecular and cellular bases of cell dormancy is a major goal in cell biology.

Fungal life cycles include the production of spores. Although being formed through largely different mechanisms, conidia (asexual spores), and ascospores and basidiospores (sexual spores) have in common to be stable dormant cell types [8,9]. These cells all show a variety of resistance to extreme conditions such as heat, desiccation [10,11], and many harsh complex environments such as insect guts [12] or immune system assaults [13,14]. Because of the increased resistance of spores to extreme conditions, sporulation is thought to be an adaptive strategy to survive changing environmental conditions [15]. In ascospores, which are produced by our model the budding yeast *Saccharomyces cerevisiae*, stress resistance is largely attributed to the thick cell wall of specific composition [16], and to the accumulation of protective compounds like trehalose or mannitol [9,17]. These protective features develop during sporulation, which is typically induced in vegetative yeast by nutrient stress. When spores are exposed to favorable conditions, germination coordinates the breaking of dormancy and the loss of these protective features, with cell-cycle progression and vegetative growth resumption. This transition involves multiple changes in cellular state [18], including the reactivation of multiple metabolic reactions. Although the precise nutrient stimuli that drive germination is dependent on ecological contexts, a carbon source such as glucose is typically an essential signal [19].

Recent studies have shown the potential complex influence of the physical properties and organization of the cytosol in dormancy and stress resistance. Cytosolic viscosity, pH, crowding, and protein phase separation have been linked to global cell adaptation across taxonomic groups. For instance, in tardigrades, desiccation resistance is mediated by intrinsically disordered proteins that form vitrified structures [20]. Seeds of the plant *Arabidopsis thaliana* sense hydration as the key trigger for their germination through phase separation of the protein Floe1 [21]. This process is a highly responsive environmental sensor since the biophysical state of Floe1 changes within minutes when water content is altered [22]. Examples of the responsiveness of the biophysics of the cell cytoplasm also come from yeast, such as *S. cerevisiae*, responding to acute stresses. Early heat shock response in yeast includes cytoplasm acidification [23], viscosity adaptation [24], and protein phase separation [25–27]. Heat shock response induces the expression of many heat shock proteins composed mainly of molecular chaperones [28] that act as a dispersal system for the heat-induced phase-separated protein condensates and promotes the rapid recovery from stress [29].

Given that budding yeast spores are inherently resistant to stresses that are known to modify many biophysical features of the cytoplasm, we hypothesize that the spore cytoplasm has biophysical properties similar to cells exposed to acute stress and that these will dynamically change during early spore germination. Here, we therefore examine the biophysical properties of dormant budding yeast ascospores and the changes that occur during dormancy breaking to unveil the molecular processes that support this critical life history cell transition. Our results reveal that dormant spore cytosol is highly rigid and acidic and that breaking of dormancy is supported by the neutralization and increased fluidity of the cytoplasm. We used mass spectrometry to examine proteome-wide protein solubility through germination. The measurements of 895 proteins revealed dynamic changes in protein solubility. We uncovered, for instance, the solubilization of several metabolic enzymes during this transition. Our results demonstrate that spores have exceptional biophysical properties and that many of the changes taking place in spores mimic what occurs in yeast experiencing stress relief. One major similarity is the implication in spore germination of a small heat shock protein, Hsp42, which is essential for normal spore activation and whose activity is regulated by its phosphorylation.

## Results and discussion

### Spores have a dense cytoplasm and display a different ultra architecture that changes during germination

Spore germination is the transition of dormant spores toward metabolically active and dividing vegetative yeast cells. Spores and vegetative yeast differ in terms of morphology and this morphology gradually changes through time. Spores are spherical and highly light refractile (Fig 1A) and darken and start growing quickly after the initiation of germination which can be induced by transferring cells to rich media. The hallmark of the completion of germination and the return to vegetative growth is bud emergence, which occurs at about 6 h after induction of germination (Fig 1A). Spores' transition from high to low refractility correlates with the decrease in optical density at 595 nm ($A_{595}$) of the pure spore culture [30], with the minimal values reached about 3 h after induction (Fig 1B). Then, subsequent growth leads to an increase of optical density. One of the adaptive features of spores is their resistance to heat. This feature is lost during germination. The quantification of heat shock resistance during germination highlights a drastic cellular transition as early as 1 h after induction, at which point resistance to thermal stress decreases and reaches levels that compare to that of vegetative yeast (Fig 1C). Taken together, these measurements define the time frame and the major time points can be used to examine the underlying cellular and molecular changes.

We obtained a more detailed view of the inner cell during germination using transmission electron microscopy (TEM, Fig 1D). Dormant spores are distinguishable by their small size and the thick spore wall, which is not seen in vegetative cells. The spore cytoplasm appears darker in TEM in comparison to a vegetative yeast, which suggests a denser cytosol (Figs 1D and 1E and S1A). Spores have a different cytoplasmic organization. This is shown by the membranous structures that look highly packed in dormant spores compared to in vegetative yeast (Fig 1D). Cells at 1 and 2 h into germination are still indistinguishable from dormant cells. Visible cytoplasmic organization changes after about 3 h of germination, which correlates with a drop in heat resistance comparable to levels seen in vegetatively growing yeast (Fig 1D). At this time point, there is a rupture of the outer spore wall and the cell starts increasing in size where the spore wall is open (Fig 1D). This size increase is accompanied with a decrease in cytoplasm density (Fig 1E). These observations suggest that transformation of the physical nature of the cytosol environment coincides with germination and return to vegetative growth.

To test the cytosol physical properties during germination, we quantified its dynamic through the examination of macromolecular motion. We expressed the reovirus non-structural protein μNS tagged with GFP as a foreign tracer particle, which has shown to be a suited probe for subcellular environment in yeast [31]. μNS self-assembles in 1 or 2 discrete particles in the yeast cytoplasm that we could detect in both spores and vegetative yeast (Figs 1F and S2). The tracking of single particles revealed their lower mobility in dormant spores compared to vegetative cells (Fig 1F). These measurements suggest that dormant spore cytoplasm is highly rigid or dense. Particle motion remained low during the first 2 h of germination, then increased gradually from hatching (3-h time point) until the end of germination (Fig 1F). At bud emergence, the motion of μNS particles is close to that measured in vegetative cells. Other experiments using μNS as tracer particles reported mean squared displacement (MSD) at 1 s lag-time in the order of $10^{-1}$ μm$^2$ [31], which is in the range of our results. While energy deprivation in yeast reduces MSD of tracer particles by less than 1 order of magnitude [31,32], we estimated that particles have motion 2 orders of magnitude lower than in vegetative yeast. These results highlight that dormant spores have an exceptionally dense and rigid cytosol. These observations are in agreement with previous work on the fungi *Talaromyces macrosporus*, where spores were found to be characterized by high viscosity [33].

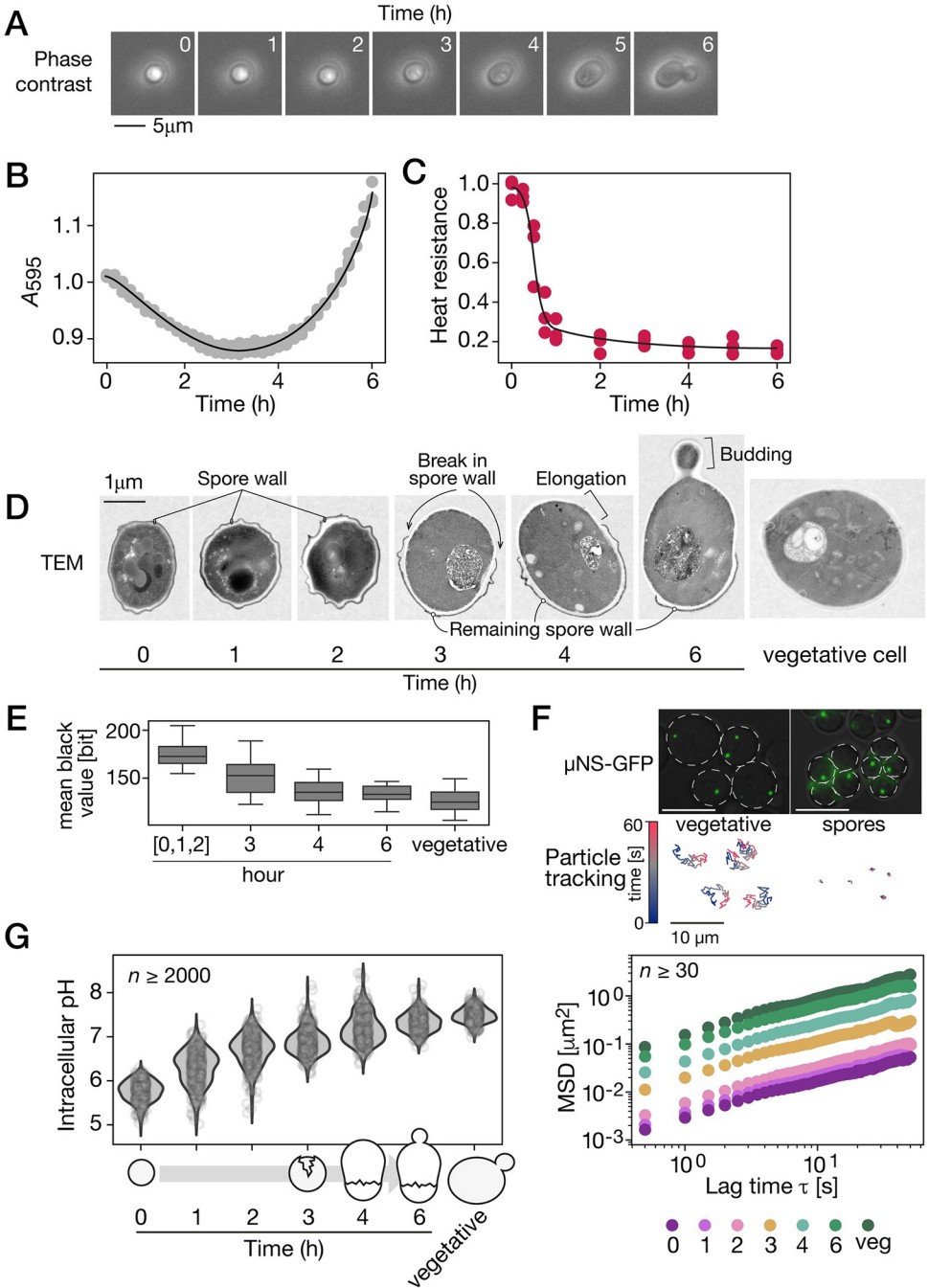

**Fig 1. The cytoplasm of dormant spores displays high rigidity and density and is an acidic environment.** (A) Phase-contrast microscopic images of ascospore (same cell followed through time) at the indicated time after exposure to rich media, which activates germination. The scale bar represents 5 μm. (B) Optical density ($A_{595}$) and (C) heat resistance of pure spore cultures through time after exposure to rich media. Heat resistance is the ratio of growth after a heat shock at 55°C for 10 min to growth without heat treatment. Experiments were performed in triplicate and values for individual replicates are shown. (D) Representative TEM images of spores at the indicated time after exposure to rich medium and of a vegetatively growing yeast cell (vegetative). Cells were prepared and stained at the same time. Imaging was performed on a single layer. The scale bar represents 1 μm. See S1A Fig for more examples. (E) Mean black level of spore cytosol at the indicated time after exposure to rich medium and of vegetative yeast. Spores at 0, 1-, and 2-h time points are merged into a single category since they are indistinguishable from one another. (F) Top, microscopic images of μNS-GFP particles in vegetative yeasts and spores. Underneath is the corresponding 1-min trajectories of the particles. Color indicates time scale. The scale bar represents 10 μm. Bottom, ensemble MSD of μNS-GFP particles in spores at the indicated time after exposure to rich medium and in vegetative cells. (G)

Intracellular pH measured at the indicated time point after germination induction and in exponentially growing cells. Measurements in at least 2,000 cells are shown at each time point. The data underlying this figure can be found in S1 Datasheet. MSD, mean squared displacement; TEM, transmission electron microscopy.

Stress response in yeast includes cytoplasm acidification that culminates with its rigidification [31], including during heat shock [23]. We therefore hypothesized that the high viscosity of the spore cytoplasm and heat shock resistance would be accompanied by a low pH that would increase during germination. To test this hypothesis, we constitutively expressed the pH biosensor superfold-pHluorin [34], in both vegetative yeast and spores after calibrating pHluorin fluorescence in vivo. We estimated pH to be around 5.9 in dormant spores, confirming previous reports [35,36]. Over the course of germination, the cytosol is gradually neutralized (Fig 1F). As soon as 1 h after exposure to rich media, median intracellular pH rises to 6.2 and it slowly increases until the end of the process ($pH_i$ = 7.3). At this point, intracellular pH gets close to that measured in vegetatively growing cells ($pH_i$ = 7.4). Previous works showed that acidification and alkylation of yeast cytosol causes reduction and increase of motility of μNS particles respectively, and that this effect happens quickly, in the scale of a few minutes [31]. However, our results show that during germination, the kinetics of change in particle mobility is delayed compared to change of intracellular pH. Although germination involves physicochemical changes related to that seen in vegetative yeast recovering from stress, they are modulated in a germination-specific manner. Our results reveal the contribution of possibly many other factors to changes in viscosity.

Altogether, these experiments show that extreme physicochemical conditions prevail in dormant spores compared to vegetative yeast, namely a highly rigid and acidic cytoplasm. These conditions are modulated during germination and return to vegetative growth. These intracellular properties that change during germination can play a critical role in cellular function and organization as they are some of the determinants of protein phase separation [24]. Protein phase separation was shown to underlie heat shock response in yeast and many other forms of stress responses during cell dormancy [37]. We therefore hypothesized that proteins could have a different solubility in spores and that the modification of physicochemical properties during germination affects their solubility in a time-dependent fashion.

## Protein solubility changes during germination

We adopted a physical separation technique similar to the one used in the context of heat shock to measure biochemical changes in protein solubility proteome-wide in budding yeast [27]. Protein sedimentation was driven by ultracentrifugation, and protein partitioning between the pellet and supernatant fractions was quantified by liquid-chromatography-coupled tandem mass spectrometry (LC-MS/MS, Fig 2A). We measured the proportion of each protein that partitioned in the pellet fraction using $P_{index}$ as a proxy for desolubilization in 3 biological replicates at 4 time points during germination and in vegetative cells. In total, we detected 24,559 unique peptides corresponding to 2,614 proteins across the experiments. We restricted our analysis to the 895 proteins with at least 2 unique peptides that were detected at every time point to measure $P_{index}$ (S3 Table). Values for these 895 proteins range from 0 to 1. Zero indicates that the protein was detected only in the supernatant, and 1 indicates that the protein was detected only in the pellet. Proteins with low $P_{index}$ are referred to as soluble proteins, while proteins with high $P_{index}$ as less soluble ones. Replicated measurements were strongly correlated (S2A Fig). Moreover, the solubility profile of endogenous proteins in the fractionated cell extracts revealed by western blot is similar to their $P_{index}$ trajectories (S3 Fig).

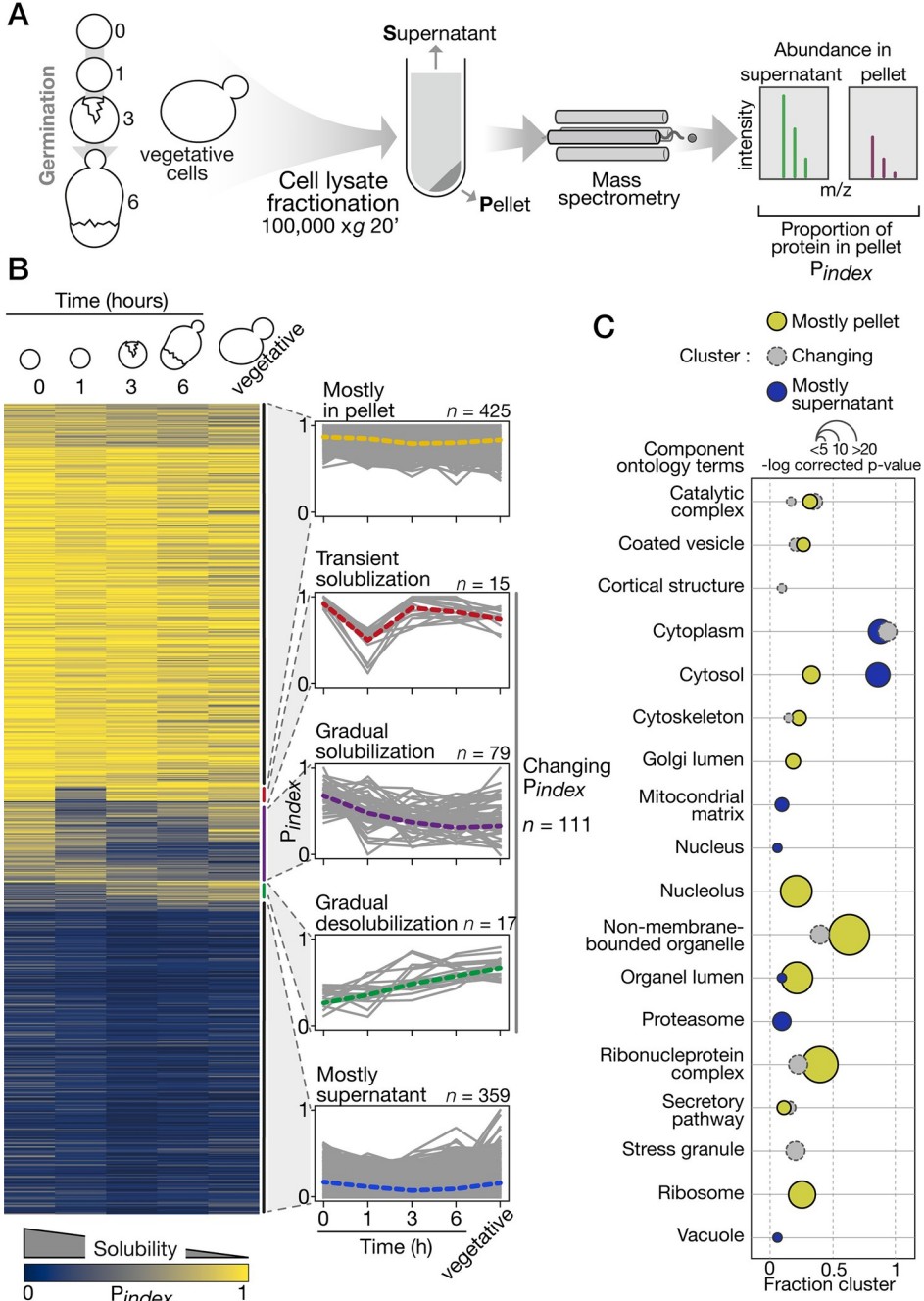

**Fig 2. Proteome-wide change in protein solubility during germination.** (A) Solubility measurement by LC-MS/MS estimates the proportion of each protein in the pellet ($P_{index}$) at each major time point sampled during germination. The experiment was performed in triplicate for all time points. (B) Right, $P_{index}$ values in the course of germination show, from top to bottom, proteins consistently found in the pellet, that transiently solubilize, that gradually solubilize, that gradually accumulate in the pellet, and that are consistently found in the supernatant. Left, individual $P_{index}$ trajectories for each cluster determined by hierarchical clustering. The dotted line is the median trajectory for each cluster. (C) GO term analysis focused on cellular component terms. Terms enriched for each cluster (Mostly in pellet, Changing $P_{index}$, and Mostly supernatant) are shown as bubble plots. Colors refer to the cluster, position on the x-axis indicates the portion of the proteins in a cluster assigned to a GO term, and size of the bubble is scaled to the -log (p-values). See S2 Fig for additional details. The data underlying this figure can be found in S2 Datasheet. GO, gene ontology; LC-MS/MS, liquid-chromatography-coupled tandem mass spectrometry.

Five typical $P_{index}$ trajectories were identified using hierarchical clustering (Fig 2B). The 2 largest clusters contain proteins that remain mostly soluble (mostly supernatant, n = 359) and mostly insoluble (mostly in pellet, n = 425). Together, they account for 87% of all proteins we considered in our analysis. This means that most of the proteins do not exhibit detectable changes in physicochemical partition during germination using our approach. However, 111 proteins showed changing $P_{index}$ trajectories divided in 3 clusters. First, 15 proteins showed a transient solubilization early in germination. These proteins predominantly partitioned in the pellet in dormant spores, while 1 h after exposure to rich media, their $P_{index}$ dropped drastically before rising again at the 3-h time point and remained insoluble until the end of germination. Another group of 17 proteins gradually desolubilize in the course of germination. They start with high solubility (low $P_{index}$) in dormant spores and gradually reach higher $P_{index}$ value at later time point in the process. Finally, 79 proteins with varying single trajectories gradually gained solubility during germination. Gene ontology (GO) analysis, using all proteins considered for our analysis as a reference set, revealed that clusters are enriched for different cell component terms (Fig 2C). While the mostly supernatant cluster appears to contain essentially cytosolic proteins, the proteins in the mostly pellet and changing $P_{index}$ clusters are assigned to various and more specific cellular components. For instance, the later clusters are both enriched for proteins in non-membrane-bounded organelles, ribonucleoprotein complexes, and cytoskeleton. There is a specific enrichment for nucleolar and ribosomal proteins in the mostly pellet cluster and a specific enrichment for stress granule proteins in the changing $P_{index}$ cluster. These results highlight the level of separation performed by our technique, which seems to separate non-membrane–bounded organelles and macromolecular complexes from the other constituents of the cytosol.

We examined the properties of proteins that associate with these changes in solubility. Proteins that change solubility are not more nor less abundant than other proteins (S2B Fig). Principal component analysis (PCA) revealed that of all the protein properties considered, propensity for condensate formation (PSAP, [38]) and score for prion-like domains prediction (PLAAC, [39]) are the ones that contribute the most to the separation of proteins in terms of $P_{index}$ (S2C Fig). Because prion-like domains can contribute to protein phase separation and tune the dynamics of biomolecular condensate [40], these results corroborate the enrichments for non-membrane–bounded organelles and macromolecular complexes we reported in the clusters mostly in pellet and changing $P_{index}$. However, insolubility does not necessarily reflect phase separation as protein solubility is also influenced by many other factors such as misfolding, formation of protein/RNA granules, or other homogeneous or heterogeneous oligomerization. Nevertheless, because propensity for condensate formation positively correlates with $P_{index}$, high $P_{index}$ estimates at least partially reflect phase separation of proteins and macromolecular assemblies. In addition, the analysis of known physical interaction among the detected proteins revealed that changes we reported in $P_{index}$ do not correlate with large interaction networks. Out of the 111 changing $P_{index}$ proteins, 6 pairs of physically interacting proteins were found (S4 Fig). This suggests that the changes in protein organization we observed likely reflect bulk changes in cytoplasm properties rather than remodeling of specific interactions.

## Many classes of proteins change solubility during germination, including metabolic enzymes

To understand the functional significance of change in $P_{index}$, we searched for GO terms enrichment in 3 clusters that display dynamic change. First, in the transient solubilization cluster, we found significant enrichment for lipid and phospholipid-binding proteins (Fig 3A). This group includes, for instance, the translation initiation factor Cdc33 and the GTP-binding

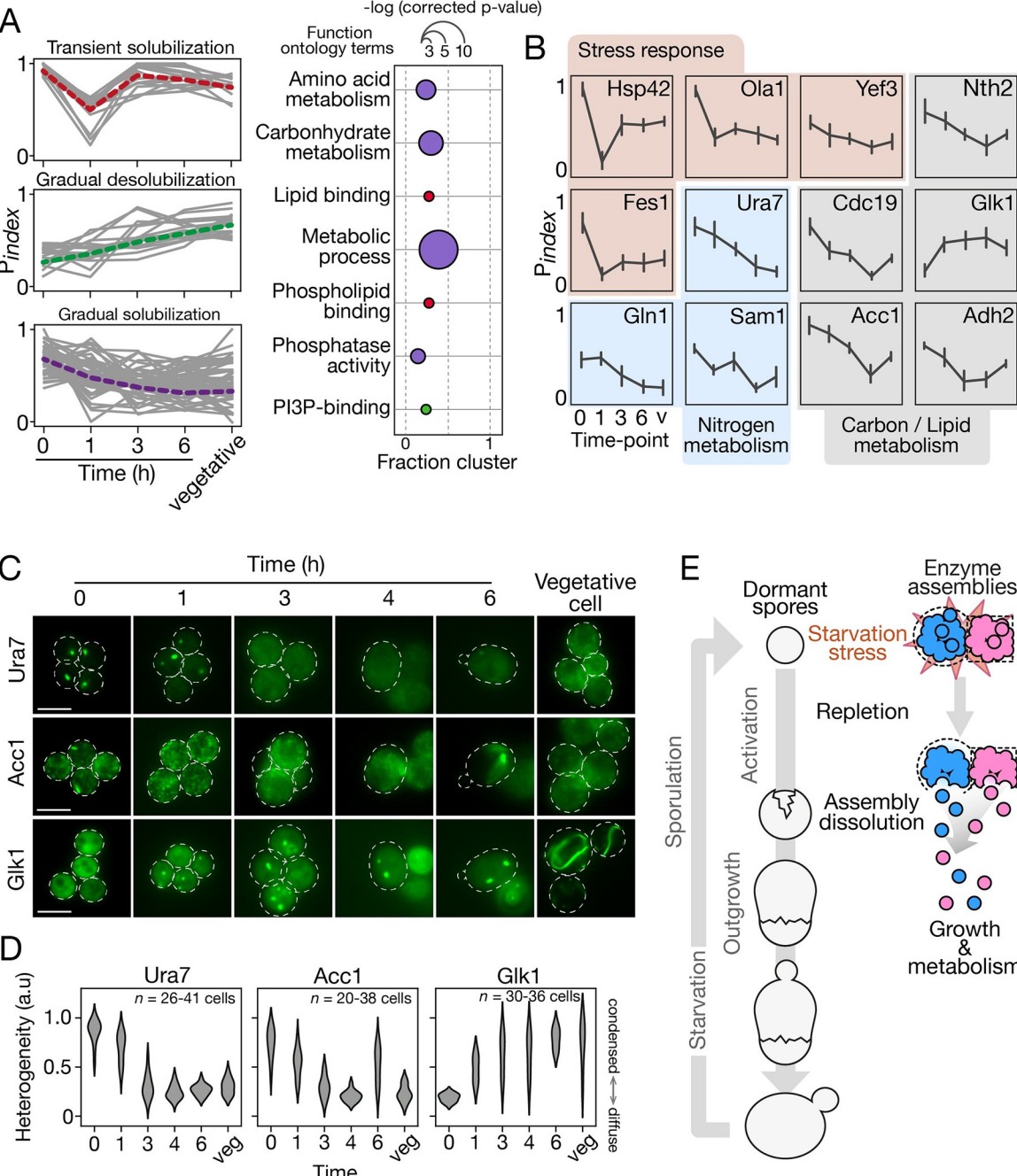

**Fig 3. Solubility changes reflect metabolism activation and mimic stress relief during germination.** (A) Enrichment for GO terms in each dynamically changing solubility cluster. Red, transiently solubilizing cluster; green, gradual desolubilization cluster; purple, gradual solubilization cluster. The position on the x-axis indicates the portion of the proteins in a cluster assigned to a GO term, and size of the bubble is scaled to the -log (p-values) from a hypergeometric test. (B) Individual $P_{index}$ trajectories for representative proteins through germination. Proteins are clustered by function; red, stress response proteins; blue, nitrogen metabolism proteins; gray, lipid and carbon metabolism proteins. Error bars represent standard deviation of 3 replicates. (C) Representative fluorescence microscopic images of spores expressing the indicated proteins tagged with GFP during germination. Top to bottom, Acetyl-CoA carboxylase Acc1 (lipid biosynthesis), CTP synthase Ura7 (pyrimidines synthesis), and Glucokinase Glk1 (glycolysis). The Glk1 foci formation and the dissolution of Acc1 and Ura7 foci in course of germination support that dormancy in spores is analogous to a stress state and germination alleviates this state. Dotted lines indicate cell contour determined by brightfield images. Scale bars represent 5 μm. (D) Measure of cellular heterogeneity (coefficient of variation) of the fluorescent proteins in spore at the indicated time after exposure to rich medium or in vegetative cells. Between 20 and 41 cells were analyzed at each time points. (E) Schematics highlighting effects on protein solubility of nutrient starvation and repletion during sporulation and germination, respectively. Pink and blue assemblies represent

assemblies of enzymes needed for growth and metabolism during dormancy, which disassemble (pink and blue circles) during germination. The data underlying this figure can be found in S3 Datasheet. GO, gene ontology.

protein Ras2 (S5C Fig). In the gradual desolubilization cluster, which includes for instance the transcription elongation factor Spt5 and the vacuolar carboxypeptidase Cps1, we found enrichment for phosphatidylinositol-3-phosphate (PI3P)-binding proteins (Fig 3A). We suspect that the modulation in solubility we detect in these clusters is a reflection of the gain of activity of many cellular pathways. For instance, gradual insolubility of the SNARE chaperone Sec18 may reflect increasing assembly of membrane-fusion complexes as vesicle transport is resumed to sustain cell growth. Finally, the gradual solubilization cluster is enriched for proteins involved in metabolic process: precisely, amino acid and carbohydrate metabolism, and protein phosphatase activity (Fig 3A), including for instance the ceramide-activated protein phosphatase Sit4 which has roles in the G1/S transition in cell cycle [41]. This may reflect the reentry of the dormant spores in the cell cycle. Among this group, we also identified the stress-related proteins Ola1 and Yef3, which are known to aggregate in response to heat stress and disaggregate during recovery [27]. The behavior of these proteins suggest that dormancy in spores shares features with stress response and that germination would correspond to stress relief.

Within the group of proteins with increasing solubility, we identified enzymes involved in carbohydrate, lipid, and nitrogen metabolisms (Fig 3B). Since nutrient starvation is the key signal that triggers sporulation, the behaviour of these metabolic enzymes that solubilize in the course of germination caught our interest. We investigated 2 of them: the CTP synthase Ura7 and the acetyl-CoA carboxylase Acc1, which are enzymes known to form high molecular weight assemblies in response to nutrient starvation [42,43]. To validate the solubility changes revealed by $P_{index}$ trajectories, we generated cells expressing either Ura7 or Acc1 fused—at their genomic locus—with GFP. Both Ura7 and Acc1 formed cytoplasmic foci in dormant spores (Fig 3C and 3D). Upon germination, Ura7-GFP and Acc1-GFP fluorescence signals changed until they became mostly diffuse in dividing cells. This behavior confirms the dissolution of the protein assemblies observed in the $P_{index}$ trajectories. In addition, we noted the opposite behavior of the glucokinase Glk1. Glk1's $P_{index}$ trajectory suggests it gains insolubility during germination (Fig 3C and 3D). Correspondingly, we found Glk1-GFP to be diffuse in dormant spores, then appears as dense assemblies in cells as soon as 1 h after exposure to rich media and until the end of germination. Glk1 was found to polymerize and form filaments during the transition from low to high sugar conditions [44]. Its behavior in germinating cells again suggests that spores remain dormant in a starved form and that breaking of dormancy implies changes of enzyme biophysics in response to nutrient repletion.

The reverse order of events between spore germination and heat stress and nutrient stress responses for some key proteins suggests a model in which dormancy in spores is analogous to a stress response state and germination corresponds to the relief of the stress state allowing return to metabolic activity and vegetative growth (Fig 3E). Spores therefore most likely borrow stress resistance strategies we observe in vegetative yeast.

## The heat shock protein Hsp42 shows dynamic solubilization and phosphorylation during germination

To further explore the regulatory mechanisms driving cellular reorganization during germination, we searched in our proteomic data for phosphorylation on tyrosines, serines, or threonines. We identified 36 phosphoproteins with a unique phosphopeptide in at least 1 time point

during germination (Fig 4A). Given that we did not perform any enrichment for phosphorylation prior to mass spectrometry, the detection of a limited number of phosphorylation was expected. These include, for instance, the topoisomerase Top1 and the transcription elongation factor Spt5. One protein was at the intersection of the phosphoproteins cluster and the changing $P_{index}$ cluster (Fig 4B), namely the small oligomeric heat shock protein (sHSP) Hsp42. The small intersection between these clusters suggests that phosphorylation is not a primary factor driving the change in protein solubility. Since stress response in vegetative yeast involves sHSP and they were recently identified as key players in the resolution of molecular assemblies that accompany heat shock [29], we focused on this protein as one of the potential regulators of protein solubilization in germination.

Hsp42 is part of the protein clusters with changing solubility during germination. Furthermore, the solubility of Hsp42 is correlated with its phosphorylation during this time period. Solubility transiently increases while abundance of its phosphorylation also transiently increases (Fig 4B). Hsp42 was shown to reversibly assemble in heterogeneous granules in a heat-induced manner, or in quiescent cells in stationary phase [45], and to function in tuning granules assembly and disassembly [46]. Remarkably, Hsp42-dependent spatial protein organization is crucial for cellular fitness, and lack of foci formation results in a significant delay when recovering from stationary phase [45]. Notably, sedimentation of Hsp42 was found to increase overtime in yeast faced with heat stress through a physical separation technique similar to ours [27], suggesting that sedimentation reflects its chaperon function. We hypothesized that the dynamic in Hsp42 sedimentation we reported reflects its function during germination. Interestingly, Hsp42 has a unique behavior among the molecular chaperones detected in our experiments (S5D Fig), which hints for an exclusive function.

To confirm the dynamic assembly and disassembly of Hsp42 during germination, we generated cells expressing Hsp42 fused to GFP. Hsp42 accumulates in cytoplasmic foci in dormant spores, which corroborates its solubility in the proteomics experiments. One hour after the induction of germination, Hsp42 has diffused, which shows the dissolution of the foci (Fig 4D). Diffused localization of Hsp42 is only transient since foci were visible at later time points during vegetative growth. Microscopic observations therefore validate the $P_{index}$ profile of Hsp42, suggesting a transient modification of the protein taking place early in germination.

The search for phosphorylation sites from the proteomics data revealed a dynamic phosphorylation site located in the N-terminal region (NTR) of Hsp42 (S223). Disorder profile of Hsp42 highlights 3 structurally distinct domains; a central structured domain that is predicted to be an alpha-crystallin domain (ACD) common to sHSP, and a long NTR and a short C-terminal region that are both predicted to be highly disordered [47]. Structure prediction of Hsp42 (Uniprot Q12329) [48] corroborates this architecture; it predicts with high confidence a beta-strand sandwich typical of ACD and predicts large unstructured parts in the N and C-terminal regions (Fig 4E). NTRs are shown to be involved in the regulation and dynamics of chaperone activity of sHSP [47,49]. These proteins are stored in an inactive form as high-order oligomers, and their activation involves phosphorylation, especially in the NTR, that drives disassembly of sHSP into smaller complexes. For instance, several phosphorylation sites on Hsp26 were found to activate chaperon activity by weakening interactions within the oligomers [50]. The phosphorylation of S223 on Hsp42 has been previously detected by mass spectrometry. The abundance of this phosphorylation was found to increase in cells following exposure to heat [51]. Hence, we hypothesized that the phosphorylation on S223 of Hsp42 is involved in this sHSP's role during the major cytoplasmic changes that take place during germination.

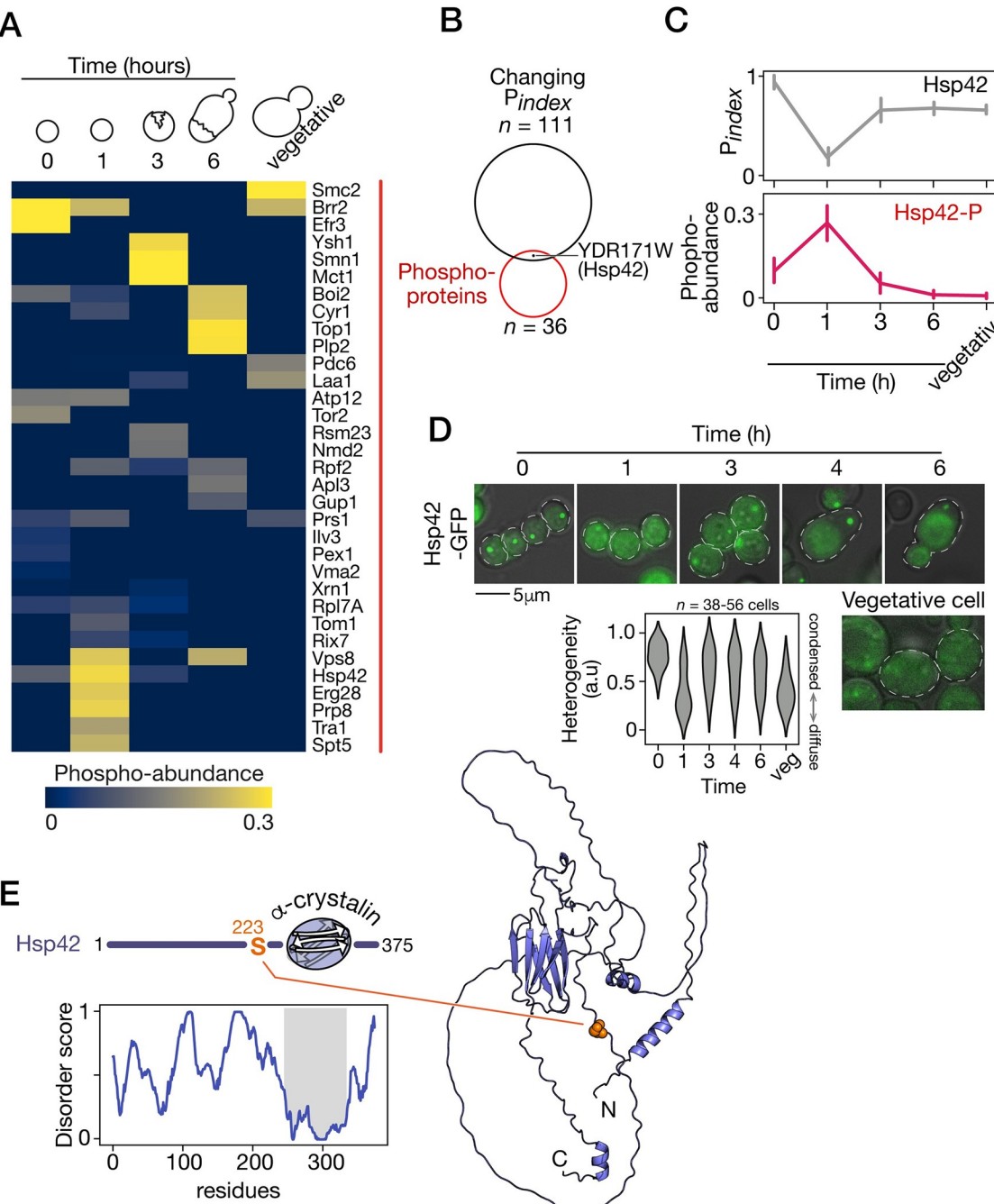

**Fig 4. Hsp42 phosphorylation at S223 is synchronized with its transient solubilization.** (A) Relative abundance of the 36 phosphoproteins to the total abundance of each protein through germination. (B) Hsp42 is phosphorylated during germination and changes solubility. See S3 Fig for additional information. (C) Hsp42 is the only protein with dynamic solubility profile during germination that correlates with its dynamic phosphorylation, here at S223. Error bars represent standard deviation of 3 replicates. (D) Representative fluorescence microscopic images of spores expressing Hsp42-GFP at the indicated time after the induction of germination. Dotted lines represent cell contour. The scale bar represents 5 μm. Plot shows the cellular Hsp42-GFP heterogeneity score in spore at the indicated time after exposure to rich medium or in vegetative cells. (E) Left, disorder profile of Hsp42, predicted by Metapredict, shows the predicted structured ACD domain, and flanking disordered N- and C-terminal region. Right, predicted Hsp42 structure. The S223 highlighted in orange is located in a disordered region. The data underlying this figure can be found in S4 Datasheet. ACD, alpha-crystallin domain.

## Hsp42 activity is crucial for normal progression of germination

We first confirmed that Hsp42 plays an important role in thermal stress protection and tested if the S223 phosphorylation may be regulating this function in vegetative yeast [46,52]. After being subjected to a heat shock, cells lacking Hsp42 (hsp42Δ::kanMX4) fail to grow as compared to WT cells, confirming thermal sensitivity (Fig 5A). A phosphomimetic mutant of Hsp42 (S223E) appears to be equally active as the WT chaperon, because expression of either protein tagged with GFP totally restores cellular heat shock resistance (S6B Fig). On the other hand, mutation of the site to a non-phosphorylatable residue (S223A) seems to impede chaperon activation or activity as revealed from the mutant phenotype. Cells expressing the Hsp42 S223A mutant show thermal stress sensitivity (Fig 5A) and this mutant fails to form large cytoplasmic foci as does the protective Hsp42 (S6A Fig). These results revealed that phosphorylation on this site is crucial for activation of Hsp42 during heat stress. We therefore tested if Hsp42 activity was crucial during germination. Optical density decrease of hsp42Δ spore cultures exposed to germination conditions is delayed compared to WT spores, suggesting a delay in germination (Fig 5C). Interestingly, microrheology revealed that hsp42Δ spores cytoplasm fluidifies in a delayed fashion compared to WT spores (Fig 5D).

Yeast adaptation to various stressor includes accumulation of compatible solute, notably the disaccharide trehalose [53]. Trehalose promotes survival by stabilizing macromolecules such as membranes and proteins [54,55]. Fungal spores, including *S. cerevisiae* ascospores, accumulate trehalose as a protection [17]. Accumulation of trehalose was found to increase yeast cytoplasmic viscosity as a homeostatic mechanism to maintain molecule diffusion rate in response to stress and energy depletion [24]. Mobilization of trehalose is essential for ascospore germination since inhibition of trehalase activity hinders germination [56,57]. Here, we investigated trehalose content in the course of germination to see if it could play a role in the changes we measured. Both WT and hsp42Δ spores accumulated a high level of trehalose. In WT spores, trehalose content dropped by >55% after 1 h in germination medium (Fig 5E), which is in line with previous works reporting a quick mobilization of trehalose content in the first hour of germination [58,59]. Contrastingly, trehalose content in hsp42Δ spores remained higher than the level in WT spores for the first 4 h of germination, and then decreased down to WT level after 8 h (Fig 5E). The delay in trehalose mobilization in hsp42Δ spore suggests that Hsp42 contributes to germination upstream of trehalose mobilization and that the major changes observed in the cytoplasm of hsp42Δ cells are at least partly caused by the dynamics of trehalose mobilization.

We tested how this delay in biophysical remodeling in hsp42Δ spores affected protein organization. In WT spores, Acc1-mCherry is diffusely localized 4 h after exposure to rich media. In contrast, in hsp42Δ spores, Acc1-mCherry remains condensed as foci (Fig 5C and 5D). These results show a direct or indirect role for Hsp42 in disassembly of Acc1 foci. Altogether, our results reveal that presence of the chaperon Hsp42 is crucial for the remodeling of intracellular conditions taking place during germination.

Expression of either WT or a phosphomimetic mutant (S223E) of Hsp42 totally rescues the germination progression in hsp42Δ spores (Fig 5A) and restores the disassembly of Acc1 foci (Fig 5F and 5G). On the other hand, spores expressing the non-phosphorylatable S223A mutant experienced a delayed germination, and in these cells, Acc1 foci failed to disassemble (Fig 5F and 5G). Altogether, activation of Hsp42 through phosphorylation on S223 appears to be crucial to disassembly of low solubility protein and germination progression. Moreover, our results revealed a surprising contribution of phosphorylation to the behavior of Hsp42 in germination. Both the non-phosphorylatable S223A or phosphomimetic S223E Hsp42 mutants were solubilized at 1-h time point, then at a later time point (4 h), the S223A mutant

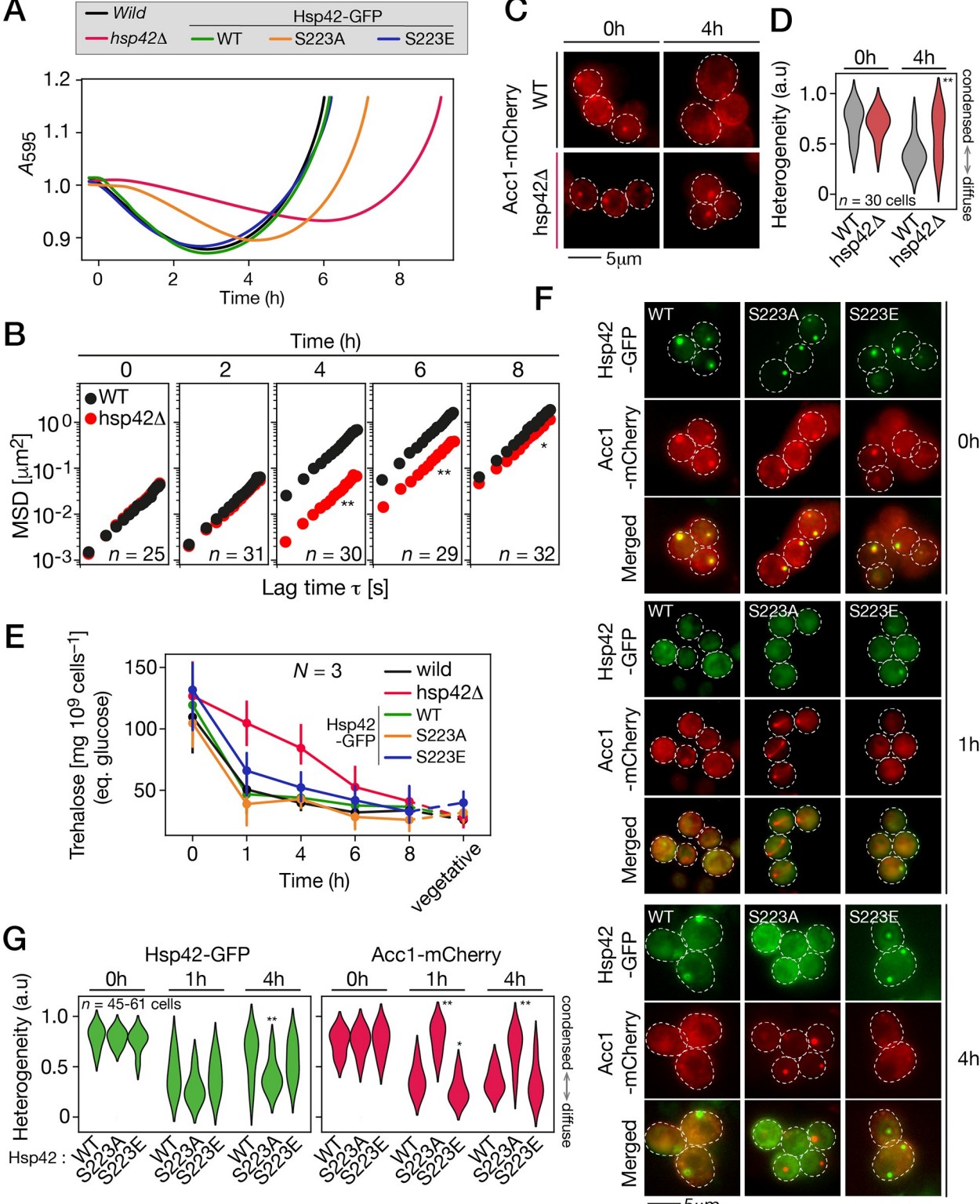

**Fig 5. Active and phosphorylated Hsp42 is required for normal germination dynamics.** (A) Optical density of pure spore cultures of the indicated strains as a function of time following exposure to germination conditions. Shown are the mean values of 3 replicates. OD drop of the hsp42Δ spores is strongly delayed, indicating that germination is inhibited or slowed down. Spores expressing S223A Hsp42 mutant show a slight delay in germination, while germination of spores expressing S223E Hsp42 mutant is indistinguishable from that of WT spores. (B) Ensemble mean

square displacement of μNS-GFP in WT and hsp42Δ spores at the indicated time after exposure to rich medium. At each time point for each strain, 25 to 35 particles, corresponding to the same number of cells, were tracked. Kruskal–Wallis test, ** indicates p-value < 0.0001, * indicates p-value < 0.01. (C) Fluorescence microscopy images of wild type (top) or hsp42Δ spores expressing Acc1-mCherry at the indicated time after exposure to germination conditions. Scale bar represents 5 μm. (D) Cellular Acc1-mCherry heterogeneity score in WT or hsp42Δ spores at the indicated time after exposure to rich medium. (E) Trehalose content (measured in equivalent of glucose concentration) in spores at the indicated time after exposure to rich medium, and in vegetative yeasts. Error bars represent standard deviation of 3 replicates. (F) Fluorescence microscopy images of spores expressing either WT, S223A, or S223E mutant Hsp42-GFP and Acc1-mCherry at the indicated time after exposure to germination conditions. Dotted lines represent cell contour. Scale bar represents 5 μm. Expression of S223A Hsp42-GFP mutant shows delay in Acc1 foci dissolution in spores. (G) Fluorescence heterogeneity score of spores expressing either WT, S223A or S223E mutant Hsp42-GFP and Acc1-mCherry at the indicated time after exposure to rich medium. Left, heterogeneity of GFP fluorescence. Right, heterogeneity of mCherry fluorescence. Kruskal–Wallis test compared to WT Hsp42-GFP, ** indicates p-value < 0.0001, * indicates p- value < 0.01. The data underlying this figure can be found in S5 Datasheet.

remained diffused in the cytoplasm while WT and S223E mutant formed foci (Fig 5F). These results refine our hypothesis about the role of this phosphorylation on Hsp42 solubilization: they suggest that some processes independent of phosphorylation on S223 are involved. Among these processes is the quick mobilization of trehalose. Our results revealed the presence of Hsp42 is crucial for trehalose mobilization, but contrastingly, it occurs regardless of the Hsp42 mutant tested (Fig 5E). Altogether, our results highlight many functions of the molecular chaperone Hsp42 for germination progression, some of which involve its phosphorylation on S223.

## Conclusion

Some dormant cells have exceptional resistance to stress. What are the biophysical conditions that underlie this property and how cells resume growth after dormancy are important questions across fields of biology. In this work, we used budding yeast ascospores to examine the biophysical properties of a dormant cytosol and its transition between dormancy and its return to vegetative growth. The spore cytosol is acidic and highly viscous, as has been observed in the context of various stresses, for instance, in yeast cells during energy depletion [31], in bacteria during metabolic arrest [60], in dry plant seed [61], and in tardigrades during desiccation [20]. The properties observed in spores may therefore represent a shared adaptive strategy for many cell types and species.

Because of the commonalities with the properties of yeast cells responding to stress, spores' cytosolic properties reflect that spores are in metabolic repression and stress response state. During germination, cells come back to an unstressed state where spore cytosol is neutralized and its viscosity decreased. We also found massive altered protein organization in dormant spores that changes along with the cytosol pH and viscosity during germination. Germination therefore shares many features with stress relief, for instance, after heat shock. One important question these observations trigger is what are the early molecular events that allow the cytosol and the solubility of many proteins to progressively change during germination. We identified Hsp42 as a key actor for the modulation of spore cytosol organization. The role of Hsp42 in dissolution of enzyme assembly during germination extends the role of chaperones in the disassembly of heat-induced protein condensate recently shown [29]. The role of heat shock proteins in response to stress in yeast may thus also be critical to the breaking of dormancy of spores, which have intrinsically high stress resistance. The dissolution of insoluble metabolic enzymes during this transition likely reflects the activation of spore metabolism as it modifies its physiology to respond to nutrients and is an adaptation to nutrient repletion [42].

We identified the phosphorylation of Hsp42 at S223 to be critical for its regulating function in protein organization. This posttranslational modification has been reported multiple times in phosphoproteomic analysis [62,63], notably in the context of heat shock where cells in

unstressed conditions show low levels of Hsp42 phosphorylation while under stress conditions they quickly accumulate phosphorylated Hsp42 [51]. Regarding these observations, the Hsp42 profile we report shows that early in germination spores exhibit stress response, and that while germination progresses, stress response is relieved. The dynamic modulation in phosphorylation of Hsp42 implies that prior signaling steps including kinase activity upstream are required for the adaptation of spore cytosol organization to nutrient repletion. Therefore, Hsp42 presumably functions in adapting spore cytosol to nutrient repletion in a similar manner as Floe1 integrates the signal of adequate hydration in *A. thaliana* seed to control germination [21]. The first event involving Hsp42 is its solubilization occurring at the breaking of dormancy. The timing of Hsp42 foci dissolution at the onset of breaking of dormancy indicated that it is likely triggered by the pathways initiating germination. While this hypothesis is not substantiated in this work, it will be worth considering the mobilization of trehalose, which functions as a macromolecule stabilizer in dormant spores, as a factor that could interact with Hsp42 to modulate the solubility of the cytoplasm. Altogether, our results expand our knowledge of the molecular factor taking part in the dissolution of protein assemblies and sheds light into the regulation of protein condensate through signaling.

Signaling and kinase activity have been previously linked with protein organization in the context of cellular stress [64]. Stress-induced phosphorylation of human Hsp27 was shown to cause its phase separation with FUS, a process that was found to prevent FUS amyloid fibril formation [65]. Phosphorylation of Hsp42 could imply the mitogen-activated protein (MAP) kinase signaling pathway, which has been reported to be involved in human Hsp27 phosphorylation [65]. Some kinases in yeast, notably cyclin-dependent kinases Cdc28 and Pho85 or MAP kinases Hog1 and Fus3, have specificities that correspond to the phosphorylation site motif of Hsp42 S223 [66]. In addition, a target of the MAP kinase Hog1, the transcription elongation factor Spt5 [67], does show a similar profile of phosphorylation during germination (Fig 4A). Connecting upstream kinases to the activity of Hsp42 will eventually allow to connect nutrient sensing of activating spores and the biophysics of spore cytoplasm.

Cell dormancy is widespread across the tree of life and is a survival strategy for many species facing harsh conditions, including pathogens [68] and cancer cells facing drug treatment [69]. By discovering what are the early events that regulate the breaking of dormancy, our work will help better understand the molecular basis of adaptation to extreme conditions and potentially help find ways to develop drugs or conditions that can potentiate existing drugs to overcome the exceptional resistance mechanisms of dormant cells.

## Methods

### Yeast strains construction and culture conditions

The yeast strains used in this study are listed in S1 Table. The genetic background for every construction is the wild diploid *Saccharomyces cerevisiae* LL13_054 [70]. This strain was chosen for its propensity to sporulate at high efficiency. For C-terminal labeling of Acc1, Ura7, Glk1, and Hsp42 with GFP at their native genomic locus (Figs 3C and 4D), GFP and Hyg resistance markers (hphNT2) were amplified from pYM25 with flanking DNA for genomic integration. For deletion of HSP42 (Fig 5, hsp42Δ), the cassette loxP-pAgTEF1-kanMX-tAgTEF1-loxP from pUG6 was amplified with the flanking DNA for replacement of the HSP42 coding sequence, leaving its promoter and terminator intact. The deletion cassette was removed by expressing the recombinase Cre on the plasmid pNatCRE. Site-directed mutagenesis on Hsp42 (S223A or S223E) was conducted by primer extension. For restoration of Hsp42 expression in hsp42Δ cells (Fig 5A and 5B and 5C and 5F), WT or mutant HSP42 coding sequences (excluding stop codon) were cloned in pYM25 upstream and in frame with GFP

using Gibson assembly. HSP42 (WT or mutant)-GFP-hphNT2 was amplified with flanking DNA for integration designed to introduce HSP42-GFP downstream of the HSP42 promoter at its native genomic locus in the hsp42Δ strain. For C-terminal labeling of Acc1 with mCherry at its native genomic locus (Fig 5E and 5F), mCherry-natNT2 was amplified from pBS35 + natNT2 plasmid with adequate flanking sequence for integration. The primers used for strain constructions are listed in S2 Table. At each step, diploid cells were sporulated, and haploid spores were dissected on selection media. Sequencing and microscopic analysis confirmed proper DNA integration. Culture from confirmed spores gave rise to homozygous diploid cells as these are homothallic spores. Competent cells were prepared and transformations performed using standard protocols [71]. Yeast were grown in YPD medium containing 1% yeast extract (Bioshop), 2% peptone (Bioshop), and 2% glucose (Bioshop) with the appropriate antibiotics for selection.

## Sporulation and germination

Sporulation was conducted on sporulation medium plates containing 1% potassium acetate, 0.1% yeast extract, 0.01% glucose, and 2% agar and spores were further purified on Percoll gradient (Sigma) as previously described [30]. Germination was induced by transferring spores to YPD. To monitor germination, fresh spores were diluted in YPD at an OD600 = 1 and optical density was measured periodically in an Infinite M Nano plate reader (Tecan) set at 30˚C.

## Heat shock resistance assay

Resistance measurements in spores during germination (Fig 1B) were conducted as described previously [30]. Briefly, freshly purified wild spores were induced to germinate in YPD medium, and at the indicated time following induction cells were sampled. Half of the cells were diluted in YPD medium, and the other half was treated at 55˚C for 10 min in a thermocycler (Eppendorf Mastercycler ProS) before being transferred to YPD. Growth curves of both treated and untreated cells were recorded in an Infinite M Nano plate reader (Tecan) set at 30˚C without shaking. Area under the curve (AUC) was measured using the Growthcurver package in R [72]. Heat resistance score was defined as the ratio of AUC of treated growth curves to AUC of untreated growth curves both obtained over the time required for untreated spore ODs to reach stationary phase. For resistance measurement of vegetative cells (Fig 5A), cells were grown overnight in YPD and diluted in YPD at OD600 of 0.1 and grown at 30˚C until they reached an OD600 of 0.4 to 0.5. Equal amounts of cell were diluted in fresh YPD medium or incubated at 50˚C for 10 min in a thermocycler prior to dilution. Growth curves of treated and control cells were recorded in a plate reader set at 30˚C.

## Phase contrast and fluorescence cell imaging

All microscopic imaging experiments were performed using eight-well glass-bottom chamber slides (Sarstedt) coated with 0.05 mg/ml concanavalin A (Millipore Sigma). For phase contrast observation of germination (Fig 1C), freshly prepared spores were induced in germination by transferring them in a chamber filled with YPD medium. Cell imaging was performed on an Apotome Observer Z1 microscope (Zeiss) equipped with LD PlnN 40×/0.6 objective (Zeiss) at the indicated time after induction in a single field. For fluorescence observation during germination, freshly prepared spores were diluted in YPD medium and incubated at 30˚C. At the indicated time after exposure to germination conditions, spores were washed in water and transferred in a chamber filled with SC medium containing 0.174% Yeast nitrogen base (BioShop), 2% glucose, and 0.5% ammonium sulfate (BioShop). For fluorescence observation on vegetative cells (Fig 5B), cells were grown in YPD at 30˚C until they reached an OD600 of 0.4

to 0.5. Cells were left untreated (control) or subjected to a heat shock at 50˚C for 10 min in a thermocycler. They were then washed in water and transferred in a chamber filled with SD medium. Fluorescence imaging was performed on an Apotome microscope equipped with a Plan-Apochromate 100×/1.4 oil objective (Zeiss). Image acquisition was performed using an AxioCam MRm camera (Zeiss). Images were analyzed using ImageJ [73].

### Fluorescence heterogeneity score

To measure cellular heterogeneity, fluorescent images were processed in ImageJ and cell peripheries were drawn by hand to analyze fluorescence signals of single cells. Heterogeneity scores are the fluorescence signal coefficient of variation, obtained by dividing the standard variation by the mean signal.

### Transmission electron microscopy

Freshly prepared spores were induced in germination in YDP at 30˚C. At the indicated time after the induction of germination, cells were harvested, washed in water, and suspended in a solution for fixation containing 2.5% glutaraldehyde, 1.5% paraformaldehyde, 0.5 mM $CaCl_2$ in 0.1M caco buffer (pH 7.2). Vegetatively growing cells in YPD (OD600 = 0.5 to 0.6) were harvested, washed in water, and suspended in this solution. Cells were fixed for 24 h at room temperature. The following steps were conducted by the microscopy platform of IBIS (Université Laval, Québec, Canada). Cells were dehydrated in an ethanol solution (30% to 100%), then embedded in epoxy resin (Epon), and 150 nm–thick sections of resin-embedded cells were prepared using an ultramicrotome (Ultracut UCT; Leica), stained with 1% (wt/vol) uranyl acetate in 70% (wt/vol) methanol for 5 min and with 0.4% lead citrate for 3 min. Samples were imaged on a JEM 1230 Transmission Electron Microscope (JOEL). Images were analyzed and processed using ImageJ.

### Molecular probes

The complete sequence of mammalian orthoreovirus 3 strain T3 nonstructural protein μNS (GeneBank MK246417.1) was kindly shared by Pr. Martin Bisaillon from Université de Sherbrooke. We cloned by Gibson assembly the whole coding sequence, minus stop codon, into pYM25 [74] to generate a fusion with yeGFP at its C-terminus. The promoter of SOD1 (nucleotides −851 to −1 relative to ATG) was cloned by Gibson assembly upstream the μNS coding sequence in pYM25. Expression of SOD1 was shown in spores and during germination [75], and expression of the molecular probe with this promoter happened at a high level in spores and during germination, which suited our experiments with this cell type. SOD1 promoter -μNS—GFP in addition to HPH markers on pYM25 were amplified as a whole with the appropriate flanking sequences for genomic integration at the URA3 locus. From all tested loci for integration (MET15, LEU2, HIS3), URA3 allowed high and uniform expression of the probes across the population, while having the least effect on sporulation and germination efficiency.

Plasmid p426MET25 containing sfpHluorin was purchased from Addgene (ID 115697). We swapped the yeGFP gene in pYM25 for sfpHluorin, and cloned the SOD1 promoter upstream the sfpHluorin coding sequence by Gibson assembly. The SOD1 promoter—sfpHluorin in addition to HPH marker on pYM25 were amplified as a whole with the appropriate flanking sequences for genomic integration at the URA3 locus.

Yeast cells with either genomic integration were selected using hygromycin resistance.

## Particle tracking and microrheology

Cells expressing μNS-GFP were transferred to an 8-well glass-bottom chamber slides (Sarstedt) coated with concanavalin A 0.05 mg/ml (Millipore Sigma) and filled with 500 μl of SC medium. Image acquisition was performed using a Perkin Elmer UltraVIEW confocal spinning disk unit attached to a Nikon Eclipse TE2000-U inverted microscope equipped with a Plan Apochromat DIC H 100×/1.4 oil objective (Nikon), and a Hamamatsu Orca Flash 4.0 LT + camera. Imaging was done at 30˚C in an environmental chamber. The software NIS-Elements (Nikon) was used for image capture. For each field, 1 brightfield and a series of fluorescence (GFP) images were taken. Cells were excited with a 488 nm laser and emission was filtered with a 530/630 nm filter. GFP time lapse images were acquired continuously at a rate of 2 frames/sec for 1 min. Images were processed using imageJ. Particle tracking was performed using the python package Trackpy ([76], http://soft-matter.github.io/trackpy/v0.5.0/). Particles were identified in microscopic images using the "locate" function. Minimal mass threshold was set at 200 to exclude spurious fluorescence signals. Trajectories were assembled from the multiple frames using the "link" function. The "imsd" and "emsd' function was used to compute mean squared displacement of individual particles and ensemble MSD, respectively. Microns per pixel was set to 10/75 and frames per second to 2.

## Intracellular pH measurements

Exponentially growing wild-type cells expressing sfpHluorin (OD = 0.3 to 0.4) in YPD medium were used for calibration curve determination as previously described [77]. Cells were washed twice in water and suspended in calibration buffer containing 50 mM NaCl, 50 mM KCl, 50 mM MES, 50 mM HEPES, 100 mM ammonium acetate, 10 mM 2- deoxyglucose, and 10 μm nigericin; pH was adjusted with HCl or KOH from 5.0 to 9.0. After 30 min incubation at room temperature, fluorescence (533 nm) of sfpHluorin following excitation at 405 and 488 nm was acquired using a Guava EasyCyte HT cytometer (EMD Millipore). The calibration curve was generated by taking the median ratio of fluorescence after excitation at 405 nm to excitation at 488 nm (405/488 ratio) at various pH. Ratios were corrected for background by subtracting the autofluorescence of unlabeled cells (WT). Points were fitted to a sigmoid (S1D Fig). Viscosity is among many parameters that affect the response of pHluorin to its environment [78]. To confirm that pHluorin sensitivity and response in the highly dense and rigid ascospores cytosol is comparable to that in vegetative cells, we performed a calibration curve in spores. Spores expressing pHluorin were incubated in a calibration buffer adjusted to various pH for 1 h at room temperature before fluorescence was analyzed by flow cytometry as for vegetative yeast. Fluorescence ratios showed that pHluorin in spores has a similar sensitivity and response compared to that in yeast cells (S1D Fig). pH measurement was performed on vegetatively growing cells (OD = 0.3 to 0.4) expressing sfpHluorin in YPD and freshly prepared spores expressing sfpHluorin at the indicated time points after exposure to rich medium. Cells were washed twice in water then suspended in a measurement buffer containing 50 mM NaCl, 50 mM KCl, 50 mM MES, 50 mM HEPES, and 100 mM ammonium acetate. After 30 min of incubation at room temperature, the median 405/488 ratio was measured by cytometry. pH values were obtained from the sigmoid function of the calibration curve.

## Protein extraction and sedimentation

Freshly purified wild-type spores at the indicated time following germination induction in YPD medium and vegetatively growing cells in YPD (OD = 0.5 to 0.6) were harvested. Cell were resuspended in 4 ml of protein buffer containing 120 mM KCl, 2 mM EDTA, 20 mM HEPES-KOH (pH 7.4), 1:500 Protease inhibitor (MiliporeSigma), 0.5 mM DTT and 1 mM

PMSF, and snap frozen as 20 μl beads, then placed in a 10 ml milling pod (Retsch) cooled in liquid nitrogen along with a 10 mm milling bead. A total of 20 milling cycles of 2 min each were performed on a Mixer Mill MM 400 (Retsch) at 30 Hz, with cooling in liquid nitrogen between cycles. Cell extracts were thawed on ice and clarified by centrifugation at 16,000 g for 10 min. Supernatant was retrieved and protein concentration was measured by BCA protein assay (Novagen, [79]). Protein concentrations were adjusted in all the samples to 800 μg/ml. Equal volume (2 ml, i.e., 1,600 μg) of cell extracts were loaded in ultracentrifuge tubes (Beckman). Samples were ultracentrifuged at 100,000 g for 30 min at 4˚C in an Optima XPN-100 ultracentrifuge (Beckman). Supernatants were kept aside as the "Supernatant" fraction. Pellets were washed twice with protein buffer, then resuspended in protein buffer + 1% SDS that corresponds to the "Pellet" fraction. Approximately 1% of total supernatant (i.e., 20 μl) and pellet fraction (i.e., 10 μl) for each cell extract was loaded on a 10% SDS-polyacrylamide gel in a loading buffer containing 0.06M Tris (pH 6.8), 0.07M SDS, 10% glycerol, 5% 2-mercaptoethanol, and 0.01% bromophenol blue. Migration was performed at 90 V until the dye front reached 1 cm into the gel. Proteins were stained with Coomassie G-250 dye, and lanes were cut out of the gel and stored in 1.5 ml microtubes before they were further processed.

## Western blot

For the detection of native proteins in fractionated cell extracts (S3 Fig), 1% of supernatant and pellet fraction for each time point in germination were loaded on a 12% SDS-polyacrylamide gel. Migration was conducted at 120 V until the dye front reach the bottom of separating gel. Proteins were then transferred to a nitrocellulose membrane (Li-Cor) and blocked for 2 h at room temperature in a blocking Buffer (Li-Cor). The following antibodies were used for detection of Bcy1, Homocitrate synthase and actin, respectively: anti-Bcy1 (yN-19, Santa Cruz Biotechnology, SC-6764), anti-homocitrate synthase (31F5, Santa Cruz Biotechnology, SC-57832), and anti-actin (Clone C4, EMD Milipore, MAB1501R). After washing in phosphate buffered saline (PBS) 1× containing 1% Tween 20, membranes were incubated with the appropriate fluorophore-conjugated antibodies (Li-Cor). Blots were then imaged on an Odyssey Imager (Li-Cor), and images were analyzed on Image Studio software (Li-Cor, v1.1).

## Mass spectrometry

In gel protein digestion was performed as previously described [80]. Gel lanes of each sample were cut into smaller pieces, destained with 40% ethanol in 30 mM ammonium bicarbonate then reduced with 10 mM DTT at 37˚C for 30 min then alkylated with 55 mM iodoacetamide at 37˚C for 30 min. The gel pieces were digested at 37˚C initially with 0.5 μg of trypsin (Promega) per sample for 6 h then additionally with 0.3 μg of trypsin overnight. The resulting peptides were extracted from gel pieces using sequential shaking in 40% acetonitrile then 100% acetonitrile, vacuum centrifuged (Vacufuge, Eppendorf) to evaporate the organic solvents and cleaned through C18 STop-And-Go-Extraction tips (StageTips, [81]), eluted in 40% acetonitrile, 0.1% formic acid, and vacuum centrifuged until complete dryness.

LC-MS/MS analysis followed [82]. The concentration of the final reconstituted sample was measured at A205 using a NanoDrop One (Thermo Fisher) to inject 250 ng into Bruker Impact II Qtof coupled to easy nLC 1200 [82]. The injection was randomized to minimize loading order bias. A single analytical column set up using IonOpticks' Aurora UHPLC column (1.6 μm C18 and 25-cm long) was used to create 90 minutes of separation from 5% to 35% buffer B for each sample.

## Data search

Resulting data were searched on MaxQuant version 1.6.17.0 [83] against sequences from verified and uncharacterized ORFs from the R64-3-1 release of the S288C genome proteome database (yeastgenome.org) and common contaminant sequences provided by the software (246 sequences) adding the following variable modifications: oxidation on methionines, acetylation on protein N-termini, acetylation on lysines, methylations on arginine, and phosphorylation on serines, threonines, and tyrosines. Fixed carbamidomethylation was set on cysteines. Default match between runs was enabled and default peptide and fragment mass tolerances (10 and 40 ppm) were set. Data were filtered to have 1% false discovery rates at peptide and protein levels. The mass spectrometry proteomics data was deposited to the ProteomeXchange Consortium via the PRIDE [84] partner repository with the dataset identifier PXD035403.

## Proteomic analysis

For the analysis of protein solubility ($P_{index}$ measurements), we considered the intensity-based absolute quantification (iBAQ, [85]) of proteins with sequence coverage of $\geq$10% with at least 2 peptides. A total of 895 proteins, for which total abundance (Supernatant + Pellet) was > 0 in each replicate at every time points of germination, were included in the analysis. $P_{index}$ of a given protein was measured as the ratio of its abundance in the pellet to its total abundance (Supernatant + Pellet). Since $P_{index}$ values across the triplicates were highly correlated (S2A Fig), we considered the mean $P_{index}$ across triplicates. $P_{index}$ values of the 895 proteins considered at each time point in germination are listed in S3 Table. Clustering of $P_{index}$ trajectories was performed in python using the Hierarchical clustering method in the Scipy package (scipy.cluster.hierarchy). Hierarchical linkage was conducted with the linkage function using the "complete" method. The clusters were then defined using the fcluster function using the "distance" criterion for discrimination.

## Protein properties

Molecular weight and isoelectric point of the 895 considered proteins were retrieved on the web-based YeastMine application (https://yeastmine.yeastgenome.org/). Total iBAQ (Supernatant + Pellet) for each of the 895 proteins in the analysis was average among the 5 time points to obtain the mean abundance. To measure, in the considered proteins, the amino acid composition predicted to form prion-like domain, we used the Prion-like amino acid composition (PLAAC) web-based application (http://plaac.wi.mit.edu/details, [39]). From this application, we considered the normalized score (NLLR) of each protein for our analysis. To predict the propensity of each of the proteins to condensate, we used the python application PSAP ([38], https://github.com/Guido497/phase-separation). This classifier scores each residue so we used the median score of each protein for further analysis. To predict the consensus disorder of each protein, we used the python application Metapredict ([86], https://github.com/idptools/metapredict), which is a neural network trained for single residue scoring. For further analysis, we used the median metapredict score for each protein. Principal component analysis (PCA) was performed using the Scikit-learn (v1.1.3) package in python. Protein properties data were first scaled using the StandardScaler function, then PCA was performed with the PCA function.

## Trehalose quantification

Trehalose content was assayed following a method by [87]. Equal amount ($10^8$) of spores at the indicated time after exposure to rich medium or exponentially growing yeast (vegetative) were

harvested and cleaned with water. Carbohydrates were then extracted through an alkaline treatment. Cells were incubated in 0.25M $Na_2CO_3$ at 95˚C for 3 h. After incubation, pH was adjusted to 5.8 by addition of 2.5 volume of freshly prepared 0.2M NaOAc. 1% of porcine trehalase (Sigma, T8778) was added to the extract, and hydrolysis of the trehalose in glucose was then allowed by an overnight incubation at 37˚C. Glucose concentration was assayed in the extracts using a commercial kit (Sigma, GAGO20). Trehalose concentration in extracts was measured as the glucose concentration after trehalase treatment, minus glucose concentration without treatment.

## Supporting information

**S1 Fig. μNS-GFP particle size and MSD during germination; calibration curve of pHluorin emission ratio, related to Fig 1.** (A) Transmission electron microscopy images of spores at the indicated time after exposure to rich medium. Scale bar represents 1 μm. (B) Size of individual particles tracked at each time point during germination and in vegetatively growing cells. At least 30 particles were tracked at each time point. (C) Mean square displacement (MSD) of individual particles tracked at the indicated time after germination induction. (D) Left, intracellular pH calibration curves determined using vegetative yeast (black) or spores (red). Although the curve using spores is slightly more basic, measurement of pH in germination (right) using either curve shows that spores are acidic and that intracellular pH increases steadily during germination. Logistic function fitted to the data from vegetative cells was used for Fig 1G. Error bars represent standard deviation of 3 replicates. The data underlying this figure can be found in S1 Data.
(TIF)

**S2 Fig. $P_{index}$ correlation between replicates, influence of protein abundance on solubility measurements, and contribution of protein properties to $P_{index}$ distribution. Related to Fig 2.** (A) $P_{index}$ values are plotted against other replicates. Pearson's correlation coefficients are indicated on each graph. For all correlations, p-value < 0.0001. (B) Mean of the absolute abundance estimated from mass spectrometry data during germination is plotted against mean $P_{index}$ values (left), or the maximal $P_{index}$ variation (right, $\Delta maxP_{index}$) of each protein. Points are colored depending on the GO function term. Pearson's correlation coefficient with the log10-transformed abundance values are shown with the corresponding p-values. (C) PCA analysis of protein properties. Protein distribution across PC1 vs. PC2 (left) and PC1 vs. PC3 (right). Dots are colored according to the mean $P_{index}$ value. Beside the graph is the vector representation indicating the strength and direction of the contribution of each variable to the distribution; sequence-based estimation of molecular weight (MW) and isoelectric point (pI); mean abundance measured from our proteomic data; prion-like amino acid composition (PLAAC) prediction score; analysis and prediction score of phase separation (PSAP); sequence-based prediction of disorder (Metapredict). The data underlying this figure can be found in S2 Data.
(TIF)

**S3 Fig. Protein sedimentation reveals different protein solubilities, related to Fig 2.** Left, the same fractionned protein extract (S, supernatant fraction; P, pellet fraction) used in MS measurements were analyzed by SDS-PAGE with antibodies that were available for yeast endogenous proteins. Shown are anti-Bcy1, anti-homocitrate synthase and anti-actin western blots, and an identically loaded gel stained by Coomassie. Protein molecular weights in the ladder (NEB# P7706) are indicated in kDa. Right, $P_{index}$ trajectories of the proteins analyzed by western blot. Homocitrate synthase isozyme Lys20 was poorly detected at 1-h time point.

Error bars represent standard deviation of 3 replicates. The data underlying this figure can be found in S3 Data, and the raw blot images can be found in S1 Raw Images.
(TIF)

**S4 Fig. Known protein–protein interaction among proteins of the changing P_index group.**
Left, interaction between the 111 Changing P_index proteins was searched through the known physical interaction database (BioGRID v4.4.216). Pairs of interacting proteins are marked in red on the heatmap with the identity of the partners on the side. Right, the 6 pairs of interacting proteins are indicated with arrows, and the P_index trajectories for each protein is shown. Error bars represent standard deviation of 3 replicates. The data underlying this figure can be found in S4 Data.
(TIF)

**S5 Fig. Phosphorylation of Hsp42 on S223, transient solubilization cluster and molecular chaperones, related to Fig 4.** (A) MS spectra example of phosphorylated S223 peptide on Hsp42. (B) Multiple sequence alignment of Hsp42 orthologs. Numbers on top refer to residue position in the *S. cerevisiae* protein. S223 is underlined in orange. Relative conservation is shown with the bars at the bottom. Only a small portion of the sequences are shown. (C) Individual P_index trajectories for each 15 proteins in the transient solubilization cluster. Error bars represent standard deviation of 3 replicates. (D) P_index trajectories of molecular chaperones detected in our experiments. Hsp26 and Ssa1 have only partial data since they are not well detected. However, these data reveal that Hsp42 has a unique sedimentation profile among molecular chaperones. Error bars represent standard deviation of 3 replicates. The data underlying this figure can be found in S5 Data.
(TIF)

**S6 Fig. Stress activation of Hsp42 through its phosphorylation on S223, related to Fig 5.**
(A) Fluorescence microscopic images of WT or mutant Hsp42-GFP expressing cells, in control conditions (top) or after a heat shock (bottom). Dotted lines represent cell contours. Scale bar represents 5 µm. Bottom, cellular Hsp42-GFP heterogeneity measure. The S223A Hsp42 mutant shows smaller and fainter aggregates in cells, and lower heterogeneity score compared to WT and phosphomimetic mutant. Heat shock at 50°C for 10 min. (B) Growth curves of vegetative cells of the indicated strains after a heat shock (right) or a mock treatment at control temperature (left). Shown are the mean values of 3 replicates. The S223A Hsp42 mutant shows intermediate heat shock resistance between the WT (and S223E mutant) and HSP42 deleted cells. This confirms that the phosphorylation of Hsp42 at this site is important for its function. The data underlying this figure can be found in S6 Data.
(TIF)

**S1 Table. Key resource table of reagents and bioinformatics tools.**
(DOCX)

**S2 Table. Primers used in this study.**
(XLSX)

**S3 Table. P_index values at indicated time point during germination and in vegetatively growing cells.**
(XLSX)

**S1 Datasheet. Data supporting Fig 1.** Each sheet contains data for the indicated figure panel.
(XLSX)

**S2 Datasheet. Data supporting Fig 2.** Each sheet contains data for the indicated figure panel.
(XLSX)

**S3 Datasheet. Data supporting Fig 3.** Each sheet contains data for the indicated figure panel.
(XLSX)

**S4 Datasheet. Data supporting Fig 4.** Each sheet contains data for the indicated figure panel.
(XLSX)

**S5 Datasheet. Data supporting Fig 5.** Each sheet contains data for the indicated figure panel.
(XLSX)

**S1 Data. Data supporting S1 Fig.** Each sheet contains data for the indicated figure panel.
(XLSX)

**S2 Data. Data supporting S2 Fig.** Each sheet contains data for the indicated figure panel.
(XLSX)

**S3 Data. Data supporting S3 Fig.** Each sheet contains data for the indicated figure panel.
(XLSX)

**S4 Data. Data supporting S4 Fig.** Each sheet contains data for the indicated figure panel.
(XLSX)

**S5 Data. Data supporting S5 Fig.** Each sheet contains data for the indicated figure panel.
(XLSX)

**S6 Data. Data supporting S6 Fig.** Each sheet contains data for the indicated figure panel.
(XLSX)

**S1 Raw Images. Raw blot images for S3 Fig.**
(PDF)

## Acknowledgments

We thank Daniel Evans-Yamamoto, David Bradley, and Alexandre K. Dubé for their comments on the manuscript. We thank Alexandre K. Dubé and Isabelle Gagnon-Arsenault for their support in the laboratory and Alexandre Bastien for his help with the microscopy experiments. We are grateful to Pr Martin Bisaillon, from Université de Sherbrooke (Canada) for providing us the μNS coding sequence.

## Author Contributions

**Conceptualization:** Samuel Plante, Leonard J. Foster, Christian R. Landry.

**Data curation:** Samuel Plante, Kyung-Mee Moon.

**Formal analysis:** Samuel Plante.

**Funding acquisition:** Leonard J. Foster, Christian R. Landry.

**Investigation:** Samuel Plante, Pascale Lemieux, Christian R. Landry.

**Methodology:** Samuel Plante, Kyung-Mee Moon.

**Project administration:** Christian R. Landry.

**Resources:** Samuel Plante, Kyung-Mee Moon.

**Supervision:** Leonard J. Foster, Christian R. Landry.

**Validation:** Samuel Plante.

**Visualization:** Samuel Plante.

**Writing – original draft:** Samuel Plante, Christian R. Landry.

**Writing – review & editing:** Samuel Plante, Leonard J. Foster, Christian R. Landry.

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
