## [Editor Report · Decision Letter 0]

24 Aug 2022

Dear Christian, 

Thank you for submitting your manuscript entitled "Exit of spore dormancy transforms the yeast cytoplasm and the solubility of its proteome" for consideration as a Research Article by PLOS Biology.

Your manuscript has now been evaluated by the PLOS Biology editorial staff, as well as by an academic editor with relevant expertise, and I'm writing to let you know that we would like to send your submission out for external peer review.

Once your full submission is complete, your paper will undergo a series of checks in preparation for peer review. After your manuscript has passed the checks it will be sent out for review. To provide the metadata for your submission, please Login to Editorial Manager (https://www.editorialmanager.com/pbiology) within two working days, i.e. by Aug 26 2022 11:59PM.

Kind regards,

Roli

Roland Roberts, PhD

Senior Editor

PLOS Biology

rroberts@plos.org

---

## [Decision Letter · Decision Letter 1]

14 Oct 2022

Dear Christian,

Thank you for your patience while your manuscript "Exit of spore dormancy transforms the yeast cytoplasm and the solubility of its proteome" was peer-reviewed at PLOS Biology. It has now been evaluated by the PLOS Biology editors, an Academic Editor with relevant expertise, and by three independent reviewers. 

You'll see that the reviewers are all broadly positive, but each has a number of concerns that must be addressed. Reviewer #1 has several requests, two of which might involve some minor experimental work. Reviewer #2 says this is “potentially exciting” but needs extra support; one of his/her requests is to attempt the perturbation of pH and/or viscosity in order to nail the causality; s/he also suggests several other ways to support the findings by orthogonal means. Reviewer #3 has a large number of semantic and presentational concerns. The Academic Editor asked me to emphasise the point from reviewer #1 about the role of trehalose.

In light of the reviews, which you will find at the end of this email, we would like to invite you to revise the work to thoroughly address the reviewers' reports.

Given the extent of revision needed, we cannot make a decision about publication until we have seen the revised manuscript and your response to the reviewers' comments. Your revised manuscript is likely to be sent for further evaluation by all or a subset of the reviewers.

**IMPORTANT - SUBMITTING YOUR REVISION**

*Re-submission Checklist*

*Published Peer Review*

*PLOS Data Policy*

*Blot and Gel Data Policy*

Sincerely,

Roli

Roland Roberts, PhD

Senior Editor

PLOS Biology

rroberts@plos.org

REVIEWERS' COMMENTS:

Reviewer #1:

[identifies himself as Sjoerd J. Seekles]

In this article the authors investigate the biophysical properties of the cytosol of dormant ascospores from S. cerevisiae and during the early stages of germination. Additionally, the authors show that protein solubility differs between these stages, and discuss the importance of Hsp42 during this transition is transcribed. The authors also show that the ascospore's viscosity is higher than those of germlings, and that this change in viscosity through germination, by tracking µNS particles. Protein solubility changes were investigated, and specifically Hsp42 is dynamic during germination in this regard. To my knowledge, no such investigation has been done so far, and its results on spore dormancy breaking significantly enhances our understanding of this cell state, which is important for many sectors. My recommendation is to accept the paper, I only have some minor suggestions which I think could strengthen the paper:

Minor suggestions:

-Throughout the paper: I agree with the authors that dormant spores are highly stress resistant. Therefore, some of the compounds the yeast cells accumulate when stressed to become more stress resistant are likely also present in high concentrations in dormant spores. There is an overlap between dormancy and stress responses, namely both cell types accumulate compounds to stabilize proteins and membranes, therefore allowing the cell to become metabolically (almost) inactive in addition to becoming stress resistant. However, this overlap does not per definition mean that a cell entering dormancy is identical to a cell responding to a stressor. Thus, although I do value the comparison, I feel the authors should be careful when calling a dormant spore a "stress response state" and germlings an "unstressed state". Instead, I would suggest altering the statements slightly, saying that the dormant state has many similarities with a stress response state.

-Page 3: In the 2nd Alinea of the introduction, the authors introduce fungal spores as stress resistant cell types of fungi in general. When discussing all fungal spores in a broad sense, I assume this includes conidia, basidiospores and other fungal spore types. However, all information given is solely focusing on yeast ascospores. I would have liked to see some information regarding other fungal spore types, like conidia and basidiospores, or otherwise a clear transition towards the focus on specifically yeast spores.

-Page 6: Here the authors investigated the internal pH levels of ascospores by quantifying the fluorescence of a pH sensitive GFP fluorophore called pHluorin. However, I have some concern regarding this methodology. As I recall, the pH sensitivity of this fluorophore is due to conformational changes of the fluorophore during pH changes (this is what "senses" the shift in pH). Is it possible that the pH sensitivity of this fluorophore is altered in a highly viscous environment? Are the conformational changes due to pH shifts still possible when the fluorophore is in this highly stress resistant cell type, where there are high concentrations of osmolytes and a high number of chaperones present, like trehalose and heat shock proteins, that stabilize the protein? It would be beneficial to the manuscript if the proper functioning of the protein in a highly viscous environment is experimentally tested, or if this has been confirmed before, referred to the paper where this has been shown.

-Page 11: The authors show that germination is delayed in hsp42Δ cells and additionally, that there is a delayed decrease in viscosity during germination of hsp42Δ cells. There have been reports that deletions of certain heat shock proteins that act as chaperones, like Hsp104, is compensated for by the cell, by accumulating more trehalose (Elliott et al. 1996). This compensation has also been suggested for Hsp42 (Haslbeck et al. 2004), although I am not aware if this relation with Hsp42 specifically has been proven. My question: is it possible that your hsp42Δ cells accumulate more trehalose than wild-type cells and that this increased concentration of trehalose causes this delayed decrease in viscosity during germination? The potential change in internal compatible solute concentrations of these hsp42Δ cells could perhaps also influence other aspects of the conclusions. Is Hsp42 a key actor for the solubilization of low solubility proteins, or could some of these findings be explained by a change in internal compatible solute concentrations? It would strengthen the paper significantly if compatible solute concentrations, or at least trehalose concentrations, are determined for these strains during germination, or otherwise if the potential role of compatible solutes is discussed in the manuscript (see next remark). 

-Page 12: One important topic I feel is lacking in the manuscript, and specifically in the discussion, is the potential role of trehalose. Trehalose accumulates in high concentration in fungal spores and is potentially one of the (if not main) reason that the spores become more viscous. Trehalose increase is also a known stress response, and therefore would perfectly fit the comparison the authors make between dormancy and stress response state. Can the authors include some relevant information on the role of trehalose in the discussion, and reflect on the potential role of trehalose in dormancy breaking in comparison to Hsp42?

-page 13: "across the three of life" > "across the tree of life"

Reviewer #2:

Review of Plante et al.

In this manuscript Plante et al. investigate the cytoplasmic dynamics of yeast spores as they germinate. This developmental transition is association with a transformation of metabolism and loss of stress resistance. The authors find that spores have a viscous and acidic cytosol. Coincident with nutrient repletion these materials properties change, including the solubility of over 100 proteins. The authors identify Hsp42 as a potential regulator of this transition and suggest that these molecular properties are important for the exceptional stress resistance of spores in this organism. In my view this is an interesting study that adds to the growing body of literature in fungi and other organisms implicating changes in cytoplasmic materials properties in dormancy. The mechanistic links to Hsp42 are potentially exciting but need additional investigation to be fully supported. I hope the authors find my suggestions below helpful in revising the work.

Major points

1. Quantification and causality. I was struck by the differences between the kinetics of the of the uNS vs. pHluorin measurements during germination in 1D vs 1F, and how these are both much slower than the complete loss of heat resistance. This made we wonder about causality. Can the authors do an experiment that artificially perturbs pH and/or viscosity? Their argument would be greatly strengthened by experiments that move us beyond correlative understanding. This approach would also help the authors to include a control that appears to be missing for the robustness of the pHluorin measurements to changes in cytoplasmic viscosity. 

2. What does the centrifugation procedure capture, and what does it exclude? I would appreciate a fuller discussion of exactly what particle sizes are captured in the centrifugation, and what is not. Following some marker proteins by immunoblot would be especially useful. I also think cell wall/membrane digestion experiments, as have been done by Simon Alberti's group with S. pombe, would be very helpful as an orthogonal approach to distinguish between the viscous model proposed and others that involve bona fide solids.

3. Including additional data in the figures. The figures are beautiful but very minimalistic. A couple of suggestions to increase information content: a) in Fig. 3A consider showing a fuller GO term analysis. Maybe as a heatmap or bubble plot. KEGG may also be useful in this respect. b) in Fig. 4A please provide the common names of the proteins. c) in Fig. 3B how do the trajectories for other representative chaperone proteins compare? I would suggest showing - in the main probably but if not certainly in the supplement - the identities and traces for all (n=15) proteins with a similar solubilization profiled to Hsp42.

4. Is the solubility change in 111 proteins due to a couple of condensates dissolving, or changes in the bulk materials properties of the cytoplasm? I would recommend some more extensive comparisons of the 111 proteins found here to those identified in other studies in yeast. Stress granules of course, but also Q bodies and other structures such as the APOD, which is notable for its sequestration of Hsp42. More broadly, I would suggest that the authors investigate how much of what they are seeing is due to general cytoplasmic properties as opposed to a specific condensate composed of Hsp42, Acc1, and perhaps many of the other proteins? One gets some suggestions that there may be more than one assembly from Fig. 3C, but I would be more convinced by additional examples and analyses to investigate this point. Even something as simple as an analysis of which of the 111 proteins with changed solubility interact with one another in systematic P-P interaction data would be very useful.

5. Quantification. Quantification and statistical tests are essential for all micrographs. Likewise, error needs to be conveyed for the mass spec data. Just a couple of examples, but this applies to most figure panels: a) foci appear in all stages of the representative images in 4D, it is thus not entirely clear that HSP42 phosphorylation state is associated with complete solubilization. Quantification of number of cells with foci, focus size, the number of foci per cell, etc. is necessary here. b) Similarly, quantification of the images in 4D, 5E, and 5F showing the number of cells with with this phenotype is essential to reach a conclusion.

6. Linking Hsp42 phosphorylation and activity to phenotype. From the data presented the correlation between the Hsp42 phosphorylation state and solubility remains a bit unclear to me. First, the authors showed an inverse correlation between the P index of HSP42 (measured by the MS peptides) and the Phosphopeptide detected (without Phospho-enrichment in the experiment) i.e., when the protein is more soluble the phosphorylation on S223 is increased.

This inverse correlation suggests an importance of the phosphorylation on S223 for the solubility of HSP42. However, it is the phospho-dead mutant (S223A) rather than the phosphomimic mutant (S223E) that is solubilized after 4h of germination in Fig. 5F. At the same time, at time 0h Hsp42(SS223E) forms assemblies, suggesting that phosphorylation may not impede their formation either. The authors need to clarify this. The quantification I suggest may help, but considering some alternative mechanistic models may also be useful, as would a similar tighter time course for this experiment that corresponds to the points chosen for the MS data (and presented in Fig. 4D). As presented, currently presented it is unclear why S223E isn't the wild-type allele. It seems better at all activities examined. 

Additionally, the authors cite Kanshin et al. (2015): "The abundance of this phosphorylation was found to increase in cells following exposure to heat" when showing that heat shocked HSP42 is found in assemblies (Fig 5B). Again, these data suggest that phosphorylation may occur even while HSP42 is in assemblies, although solubility and focus formation may not be correlated given the diffraction limit in these experiments. The authors may want to test whether Hsp42 also becomes insoluble during heat shock using their centrifugation protocol. 

Minor points:

7. I would suggest adding some annotation in Fig 1D to guide the reader

9. The authors refer to the scheme of the LC-MS/MS in page 7 as Figure 1A instead of Figure 2A

8. It may be worth more explicitly making the point that the changes in solubility are mostly *not* about phosphorylation given the data in Fig. 4B

Reviewer #3:

In the manuscript entitled, "Exit of spore dormancy transforms the yeast cytoplasm and the solubility of its proteome," the authors evaluated spores of the budding yeast S. cerevisiae undergoing germination using biophysical, proteomic, cell biological, and genetic approaches and identified substantial changes in the spore cytosol as they break dormancy. They discovered that the spore cytosol is viscous and acidic and harbors protein assemblies that disperse and/or nucleate over time. These dynamic changes reflected changes in protein solubility consistent with the changes seen in S. cerevisiae stress responses. This led to the discovery that Hsp42 is a key regulator of the cytosolic transition whose activity was governed by its phosphorylation state. These findings support a model in which the spore state is a "stressed state" in terms of cytosolic properties and organization, and the breaking of spore dormancy leads to "stress relief" as spores transition into yeast and initiate vegetative growth. The data are compelling, and the findings are interesting; however, there are several points that need to be addressed with respect to communicating the work and interpreting the findings.

Major Points:

1) There is an overall difficulty with communication of the data because of the vocabulary used in the "yeast" field. The current presentation is confusing and needs to have more specificity/clarity if the work is going to appeal to a broad readership. It begins with the title:

"Exit of spore dormancy transforms the yeast cytoplasm and the solubility of its 

proteome"

Does "yeast" in this case mean S. cerevisiae? Or does it mean the cell type "yeast?" If it means the budding yeast S. cerevisiae, then that needs to be indicated or somehow clarified. If it means the cell type known as yeast, then the title is unclear because exiting spore dormancy transforms the spore cytoplasm to facilitate the transition into a yeast. 

Removing "yeast cytoplasm" from the title would be helpful. Suggestions to consider:

Exit of dormancy by fungal spores transforms the cytoplasm and the solubility of the proteome

Exit of dormancy by spores of S. cerevisiae transforms the cytoplasm and the solubility of its proteome

Breaking spore dormancy in S. cerevisiae transforms the cytoplasm and the solubility of its proteome

Breaking dormancy in spores of budding yeast transforms the cytoplasm and the solubility of its proteome

2) Along these same lines, the manuscript is very confusing with respect to the natures and states of the cell types in question. It is sometimes difficult to discern whether the population being discussed is dormant spores, germinating spores, germinated spores, vegetatively growing yeast, or quiescent yeast. It might seem like semantics, but in this case, it is critical to have clarity on cell type and growth state so that the data can be interpreted correctly. 

a. Part of the challenge may stem from habits in the Saccharomyces field, which tends to refer to "spores and cells" or "spores and yeast cells," and this is extremely confounding because spores are cells. A workable solution is to simply refer to the two cell types in question (spores and yeast) and avoid the use of the word "cell" altogether whenever possible. One can then use a descriptor to modify the type of spore or yeast being discussed. For example, "germinating spore" or "replicating yeast." Because there are relatively quiescent states for yeast (such as stationary phase), the state of the yeast should be made clear to readers (because not all yeast are growing vegetatively). It is clear that the authors have attempted to differentiate among the different cell types and states in many instances, but this should be revisited in the text and fully addressed with consistent labels and descriptors. It doesn't really help to use the specific term "spores" and the general term "vegetative cells" when making comparisons. "Vegetative cells" has different meanings across systems from bacteria to multicellular organisms, so it would be helpful to use more specificity in this manuscript. Replace "vegetative cells" with "yeast" or "vegetative yeast" or "growing yeast" as needed. 

3) Comments on Abstract: is "Yeasts" is unclear. Not all yeasts produce spores. "Many yeasts produce . . ." followed by "We show that spores of S. cerevisiae exhibit . . ." would be more clear.

4) Comments on Introduction 

a. paragraph at the top of page 4 needs to make it clear that "yeast" in this case is S. cerevisiae. ". . . cell cytoplasm also come from yeast, such as S. cerevisiae, responding to acute stresses. . ."

b. last paragraph needs to make it clear what system and spore type is being used. "Here, we therefore examine the biophysical properties of budding yeast ascospores and the changes that occur during the breaking of dormancy to unveil the molecular processes that support this critical life-history cellular transition. 

5) Comments on Figure 1 data and results

a. Legend title - change "cells" to "spores" 

b. 1A: indicate that the germinating spore is the same one over the time course (not representative images)

c. 1B: p. 5 in reference to Figure 1B. It would be helpful comment on/speculate about why the OD595 increases after 3 hours of germination. Does refractility increase? Or is this due to an increase in optical density due to budding?

d. 1D: Apparent density in TEM can be heavily influenced by the process of fixation. Were all of these images selected from samples prepared, fixed, and imaged at the same time? Please indicate - particularly if a comparison is going to be made across images. 

e. 1D: While the text indicates that "the spore cytoplasm appears darker in TEM in comparison to a vegetative cell, which suggests a denser cytosol. . .," this appears to be the case in the images in 1D but not the case in S1A. A darker spore cytoplasm is not supported by the images shown. 

f. Is there a method to quantitate the apparent density per area? If spores getting bigger over time, one might expect the apparent density to decrease simply due to the same electron density being spread over a larger area. Is there anything in these images to counter that argument?

g. The text indicates that "Spores have a different cell organization." A clearer description would be to refer to "cytoplasmic organization." 

h. The authors also need to take care to refer to visible features changing as correlating with other properties - not as a consequence or cause of. For example, "Visible cytoplasmic organization changed after about three hours of germination, which correlated with a drop in heat resistance comparable to levels seen in vegetatively growing yeast."

i. It is not clear what the authors are intending to communicate with the sentence, "These observations suggest that the spore cytosol organization is timely modulated in the course of germination and return to vegetative growth." Perhaps -- Cytosolic reorganization temporally coincides with a return to vegetative growth?

j. It is difficult to argue that the viscosity experiments in 1E and 1F are "validating" the observation that cytoplasmic density is decreasing, when the evidence for decreasing density during germination is an observation based on TEM. It would be better to present the TEM data as suggestive, form a hypothesis, and present the viscosity and pH experiments as tests of the hypothesis. 

k. It is important that the authors clarify what was known about the behavior of µNS-GFP prior to these studies, what they showed for their specific strains, and what is newly shown in this manuscript. 

l. The reference to Figure S2 on the bottom of page 5 appears to be incorrect. Where are the µNS-GFP single molecule tracking data?? Relevant data appear to be in Figure S1B and S1C, but there isn't enough information to suss out the progression of data collection and corrections for size effect. 

m. It is unclear what particle size difference the authors are referring to at the top of page 6 (and apparently visible in Figure S1A). Are there µNS particles shown somewhere? No information in the legend for S1A.

n. The µNS-GFP measurements shown do not confirm "what we see in the TEM images." They are simply consistent with high cytoplasmic density in dormant spores and lower cytoplasmic in germinating spores with the lowest cytoplasmic density occurring in growing yeast. 

o. p. 6 first paragraph last sentence should read, "These observations are in the same range of viscosity as ascospores of the dimorphic fungus Talaromyces macrosporus, which are characterized by unusually high cytoplasmic viscosity.

p. p. 6 last paragraph - The intent of the sentence, "These conditions are timely modulated during the germination and return to vegetative growth" is not at all clear. The phrase "timely modulated" has no meaning. 

6) Comments on Figure 2 data and results

a. Reference to Figure 1A at the top of page 7 is incorrect. Should be Figure 2A. 

b. p. 7 first paragraph "Values for these 895 proteins range from 0 to 1. Zero indicates that the protein was detected only in the supernatant, and 1 indicates that the protein was detected only in the pellet." 

c. 2B: The authors indicate that LC-MS/MS was carried out on fractions from "5 time points during germination." This is incorrect according to their data. There were four times points during germination (0, 1, 3, and 6 hours) and one sample of vegetatively growing yeast. This needs clarified in the text and figure legend (unless the last timepoint is a culture of germinated spores that were left to grow as a yeast culture. In that case, it should be referred to as its timepoint of harvest after initiation of germination).

d. 2B. The authors need to indicate the identities of the genes in each of the groups shown. Table S4 could easily house this information. As it stands, it is difficult to interpret the data based on the handful of examples provided in Figure 3. 

e. 2B: It would be helpful if the groups in the figure were referred to with the same nomenclature in the Figure Legend. 

f. 2C: The image in Figure 2C appears to be identical to the one shown on the left-hand side of Figure S2C. Was Figure S2C intended to plot PC2 against PC3?

7) Comments on Figure 3 data and results

a. On Page 8 in the first half of the second paragraph the logic used by the authors is not clear. They indicate that there was no GO term enrichment for integral membrane proteins, so there must have been little membrane contamination in their preparations. Given that the preps were run on polyacrylamide gels prior to protein extraction, this seems like it should be the expectation. Perhaps they did not expect to detect lipid binding proteins and PI3P-binding proteins in the absence of membranes? Their explanation for the presence of these proteins is that there was a "gain of activity of many cellular pathways," which seems obvious, given that spores are germinating and becoming active. A more clear explanation of the expectations and outcomes here is warranted. It would also be useful to indicate explicitly why a protein like Sec18 (and its 16 companions) would show gradual insolubility when there is likely more vesicle trafficking activity. 

b. p. 8 second paragraph ". . . Ola1 and Yef3, which are known to aggregate in response to heat stress and disaggregate during recovery . . ."

c. Figure 3C: The presentations in the figure and the legend make these data confusing. (The text is more clear.) The legend states that Acc1 and Ura7 have similar behaviors as "they both condense upon stress." What's shown in the microscopy in Figure 3C is that they decondense during germination - supporting the idea that spores represent a stressed state that is relieved during germination. The legend needs to describe the discovery in this figure - not what was already known. Same for Glk1. In addition, the cartoon images shown above and below the microscopic images do not assist the reader. The images are not explained in the legend, and they are identical except for the addition of the word "relief." It is unclear what the little dots represent, what the green glob represents, and how they relate to the cells below and above. Is the dotted line representative of a cell outline? It seems like the top one is backwards, if the idea is to convey that germination likely represents stress relief. 

d. Figure 3D: The schematic of the model needs to be described more clearly in the figure legend and emphasize that the pink and blue assemblies represent only the assemblies of enzymes that are needed for growth and metabolism. It should be noted that the enzyme assemblies for proteins no longer needed have the opposite behavior. (And the model is based on the behaviors of only 3 proteins.)

8) Comments on Figure 4 data and results

a. The authors need to make it more clear what was known prior to this study and what they discovered here. They need to be conservative in their interpretations of the phospho-proteome data because of the limited (although expected) number of peptides detected. Given that Hsp42 was the only protein with the transient solubilization profile and a phosphoprotein hit, it is important that the authors provide the identities of the other 14 proteins in the transient solubilization group so that they can be evaluated.

9) Comments on Figure 5 data and results

a. The explanation for the rationale and results for Figure 5 are not clear. The top paragraph of p. 11 needs to establish what was known previously, convey the hypothesis that the authors were testing, and explain their new discoveries. 

b. The legend to Figure 5 needs to be written with proper genotypes.

Additional recommended copy edits for clarity:

In the Results section, please use the past tense for current discoveries and present tense for previously published data.

p. 3 paragraph 2 ". . . germination coordinates the breaking of dormancy . . ." 

p. 4 paragraph 2 ". . . biophysical properties of dormant budding yeast spores and the changes that occur during dormancy breaking to unveil the molecular processes that support . . .

p. 4 paragraph 2 ". . . changes taking place in spores mimic what occurs in yeast experiencing stress relief . . ."

p. 4 paragraph 2 the use of the word "implication" generally requires referral to a specific process or event. ". . . is the implication of a small heat shock protein, Hsp42 in spore germination, which is essential for . . ."

p. 5 paragraph 1 "Spores' transitions from high to low refractility correlate with a decrease in . . ."

p. 5 paragraph 1 "Taken together, these measurements . . . major time points that can be used to . . ."

p. 6 paragraph 2 "To test this hypothesis, we constitutively expressed the pH . . ."

p. 8 top paragraph 1 The use of "for instance" throughout the manuscript is unnecessary. 

p. 8 bottom paragraph change "comportment" to "behavior"

Figure 3 legend: change "comportement" to "behavior"

---

## [Decision Letter · Decision Letter 2]

15 Feb 2023

Dear Christian,

Thank you for your patience while we considered your revised manuscript "Breaking dormancy in spores of budding yeast transforms its cytoplasm and the solubility of its proteome" for publication as a Research Article at PLOS Biology. This revised version of your manuscript has been evaluated by the PLOS Biology editors, the Academic Editor, and two of the original reviewers.

Based on the reviews, we are likely to accept this manuscript for publication, provided you satisfactorily address the following data and other policy-related requests.

a) For clarity, please could you change your title slightly to "Breaking spore dormancy in budding yeast transforms the cytoplasm and the solubility of the proteome"?

b) Please address my Data Policy requests below; specifically, we need you to supply the numerical values underlying Figs 1BCEFG, 2BC, 3ABD, 4ACD, 5ABDEG, S1BCD, S2ABC, S3, S4, S5CD, S6AB, either as a supplementary data file or as a permanent DOI’d deposition.

c) Please cite the location of the data clearly in all relevant main and supplementary Figure legends, e.g. “The data underlying this Figure can be found in S1 Data” or “The data underlying this Figure can be found in https://doi.org/XXXX”

We expect to receive your revised manuscript within two weeks. 

*Published Peer Review History*

*Press*

Sincerely,

Roli

Roland Roberts, PhD

Senior Editor,

rroberts@plos.org,

PLOS Biology

DATA POLICY:

Regardless of the method selected, please ensure that you provide the individual numerical values that underlie the summary data displayed in the following figure panels as they are essential for readers to assess your analysis and to reproduce it: Figs 1BCEFG, 2BC, 3ABD, 4ACD, 5ABDEG, S1BCD, S2ABC, S3, S4, S5CD, S6AB. NOTE: the numerical data provided should include all replicates AND the way in which the plotted mean and errors were derived (it should not present only the mean/average values).

SPECIES INDICATED IN THE ABSTRACT? 

- Please note that per journal policy, the model system/species studied should be clearly stated in the abstract of your manuscript. 

We require the original, uncropped and minimally adjusted images supporting all blot and gel results reported in an article's figures or Supporting Information files. We will require these files before a manuscript can be accepted so please prepare and upload them now. Please carefully read our guidelines for how to prepare and upload this data: https://journals.plos.org/plosbiology/s/figures#loc-blot-and-gel-reporting-requirements

DATA NOT SHOWN?

REVIEWERS' COMMENTS:

Reviewer #1:

[identifies himself as Sjoerd J. Seekles]

Plante et al. have addressed all concerns raised by me and the other authors thoroughly and I accept the manuscript in its current state. I think the extra experiments and subsequent information now shown on increased trehalose content in ∆hsp42 cells is a crucial part for our understanding of the mechanisms behind dormancy breaking and changes to proteome solubility during ascospore germination. The role of Hsp42 described here is novel and interesting, and will significantly contribute to our understanding of these before mentioned processes. I would like to commend the authors on their revision. It is difficult to untangle the effects of increased trehalose levels during early stages of germination from the effects of Hsp42 inactivation/removal, and this point of contention has been adequately addressed in the Discussion section of the current manuscript.

Reviewer #2:

This is an exciting paper and I think the authors did an excellent job responding to my comments and those of the other reviewers. In particular the more nuanced discussion of the role of trehalose and Hsp42 phosphorylation has really improved the study, along with the protein-protein interaction analysis. I recommend publishing without any further delay.

---

## [Editor Report · Decision Letter 3]

21 Feb 2023

Dear Christian,

Thank you for the submission of your revised Research Article "Breaking spore dormancy in budding yeast transforms the cytoplasm and the solubility of the proteome" for publication in PLOS Biology. On behalf of my colleagues and the Academic Editor, Joseph Heitman, I'm pleased to say that we can in principle accept your manuscript for publication, provided you address any remaining formatting and reporting issues. These will be detailed in an email you should receive within 2-3 business days from our colleagues in the journal operations team; no action is required from you until then. Please note that we will not be able to formally accept your manuscript and schedule it for publication until you have completed any requested changes.

Sincerely, 

Roli

Senior Editor

PLOS Biology

rroberts@plos.org